# Benchmark of Machine Learning Force Fields for Semiconductor Simulations: Datasets, Metrics, and Comparative Analysis

**Geonu Kim**[1][*]  **Byunggook Na**[1][*]  **Gunhee Kim**[1][†]  **Hyuntae Cho**[1]  **Seung-Jin Kang**[1]
**Hee Sun Lee**[1]  **Saerom Choi**[1]  **Heejae Kim**[1]  **Seungwon Lee**[1]  **Yongdeok Kim**[1][†]
[1]Samsung Advanced Institute of Technology (SAIT)
{geonu.kim, byunggook.na, ghij.kim, robert.cho, sj1222.kang}@samsung.com
{heesun88.lee, sincere.choi, heejaeee.kim, seungw.lee, yd.mlg.kim}@samsung.com

## Abstract

As semiconductor devices become miniaturized and their structures become more complex, there is a growing need for large-scale atomic-level simulations as a less costly alternative to the trial-and-error approach during development. Although machine learning force fields (MLFFs) can meet the accuracy and scale requirements for such simulations, there are no open-access benchmarks for semiconductor materials. Hence, this study presents a comprehensive benchmark suite that consists of two semiconductor material datasets and ten MLFF models with six evaluation metrics. We select two important semiconductor thin-film materials *silicon nitride* and *hafnium oxide*, and generate their datasets using computationally expensive density functional theory simulations under various scenarios at a cost of 2.6k GPU days. Additionally, we provide a variety of architectures as baselines: descriptor-based fully connected neural networks and graph neural networks with rotational invariant or equivariant features. We assess not only the accuracy of energy and force predictions but also five additional simulation indicators to determine the practical applicability of MLFF models in molecular dynamics simulations. To facilitate further research, our benchmark suite is available at https://github.com/SAITPublic/MLFF-Framework.

## 1   Introduction

The evolution of semiconductor devices has been guided by two main factors: miniaturization and sophisticated design implementation [1, 2, 3]. Understanding complex device structures and fabrication processes requires high-resolution metrology methods, such as scanning electron microscopy and transmission electron microscopy [3, 4]. However, these instruments have limitations in terms of providing a restricted view, requiring substantial time and cost for analysis. In this context, large-scale atomic simulations are becoming increasingly important as a viable alternative approach that is capable of accurately capturing device-level dynamics. Although ab initio simulations based on the density functional theory (DFT) can provide accurate atomic simulations, substantial computational demands make them unsuitable for large-scale simulations [5, 6, 7]. As a practical substitute, molecular dynamics (MD) simulations combined with machine learning force fields (MLFFs) have gained significant attention. MLFFs aim to achieve a precision comparable to that of DFT-based simulations and relieve their computational costs.

---

[*]Equal contribution.
[†]Co-corresponding Author.

37th Conference on Neural Information Processing Systems (NeurIPS 2023) Track on Datasets and Benchmarks.

Obtaining reliable datasets and benchmarks for studying semiconductors in the condensed phase presents significant challenges, which hinders the development of accurate MLFF models. Generating precise datasets for condensed-phase materials is challenging because of the complex structures, dynamics, and large number of atoms involved. To foster the development of MLFFs in the field, we introduce two new datasets, which are specifically designed for **s**emiconductor **a**dvanced **m**aterials **d**iscovery, called SAMD23 datasets: silicon nitride (SiN) and hafnium oxide (HfO). We conducted DFT simulations under various conditions, including initial structures, stoichiometry, temperature, strain, and defects, resulting in a cost of 2.6k GPU days.

Although we built datasets with broad coverage, dynamic simulations exhibit an enormously wide range of atomic configurations with high degrees of freedom, indicating that attempting to collect all these configurations through computationally expensive ab initio simulations is not feasible in practice. Hence, MLFF models must be capable of extrapolation, which enables them to yield reliable predictions of configurations that are absent from the training dataset. Generally, to assess the extrapolation capability, an evaluation of the energy and force on out-of-distribution (OOD) test datasets, in addition to in-distribution (ID) sets, is employed. However, the energy and force errors may not be sufficient to account for the simulation behavior [8, 9]. Thus, we additionally provide five simulation indicators and correspondingly prepare material structures for both ID and OOD sets, facilitating a comprehensive comparison of the MLFF models.

Moreover, we offer a consolidated framework that streamlines model development, training, and evaluation processes into a unified platform. We curated diverse models that utilize hand-crafted features as atomic representations or employ graph neural networks as feature extractors. For the benchmark, 10 MLFF models were trained and evaluated using the framework, suggesting a reliable model selection policy based on the simulation indicators. Based on a comparative analysis of the training models with various hyperparameters, we suggest a baseline training recipe.

This paper makes the following contributions:

- We introduce SAMD23, two new MLFF benchmark datasets that reflect semiconductor simulations of SiN and HfO under various scenarios.

- We provide a framework to facilitate model development, and present benchmarks for SiN and HfO, along with five simulation indicators to assess the prediction performance in simulations and the extrapolation capability.

- We suggest a baseline training recipe and model selection policy to employ the model for simulations by performing a comparative analysis of 10 MLFF models.

## 2 Background and Related Work

**Machine Learning Force Fields.** Given a snapshot of $n$ atoms, consisting of atomic numbers $\mathbf{Z} \in \mathbb{Z}^n$ and atom positions $\mathbf{r} \in \mathbb{R}^{n \times 3}$, an MLFF model approximates the potential energy $E \in \mathbb{R}$ and forces $\mathbf{F} \in \mathbb{R}^{n \times 3}$ (see Figure B.1). MLFF models can utilize hand-crafted representations of local atomic environments, called descriptors obtained by experts in physics and material science research fields, or automatically represent the local atomic environments by adopting deep neural networks. The former method combines descriptors with different learning algorithms, such as linear regression [10], kernel methods [11, 12], and neural networks [13, 14, 15]. Recently, the latter method, that is, graph neural networks (GNNs) have shown remarkable results in predicting energy and forces by learning the features of atoms and their connectivity in an end-to-end manner instead of using descriptors. While some GNNs [16, 17, 18, 19, 20] rely on rotational invariant features such as interatomic distances and angles, others have rotational equivariant features derived from spherical harmonics [21, 22, 23, 24, 25].

**MLFF Benchmark.** Numerous MLFF benchmarks, including ANI [26], rMD17 [27], COLL [18], and 3BPA [28], have been publicly released for the application of MLFF to molecular structures. These datasets, which mainly comprise molecular structures with a small number of atoms, have been extensively utilized in the development of various MLFF models. The open catalyst 20 (OC20) [29] and OC22 [30] datasets were generated through simulations of the relaxation of a molecule on bulk surfaces. OC20 primarily investigates the relaxation behavior of diverse adsorbates on various catalyst surfaces. OC22, an enhancement of OC20, focuses oxygen evolution reactions on oxide electrocatalysts. While both OC datasets predominantly focus on surface reactions, they are less

Table 1: Summary of the semiconductor datasets. $N_{\text{cond}}$ represents the total number of simulation conditions. $N_{\text{train}}$, $N_{\text{valid}}$, and $N_{\text{test}}$ indicate the number of snapshots contained in train, valid, and test sets, respectively. The generation cost is measured by GPU hours using Nvidia V100.

| | Element | $N_{\text{cond}}$ | Snapshots | Datapoints | $N_{\text{train}}$ | $N_{\text{valid}}$ | $N_{\text{test}}$ | Cost (h) |
|---|---|---|---|---|---|---|---|---|
| SiN | $Si_m N_n$ | 92 | 76,213 | 4,397,744 | 20,315 | 2,542 | 2,585 | 29,824 |
| | Si | 14 | 6,250 | 291,600 | 1,663 | 212 | 213 | 2,963 |
| | N | 4 | 2,000 | 128,000 | 532 | 68 | 68 | 3,769 |
| | Total | 110 | 84,463 | 4,817,344 | 22,510 | 2,822 | 2,866 | 36,556 |
| $SiN^{OOD}$ | $Si_m N_n$ | 3 | 3,700 | 388,500 | - | - | 1,235 | 1,166 |
| HfO | $Hf O_2$ | 60 | 160,000 | 15,360,000 | 27,960 | 3,510 | 3,510 | 19,341 |
| $HfO^{OOD}$ | $Hf_m O_n$ | 12 | 32,000 | 3,072,000 | - | - | 6,996 | 4,182 |

optimal for semiconductor studies, which necessitate an understanding of the phenomena inside solids or at the interface between two solids. In 2023, Fu *et al.* [9] introduced a benchmark for simulation stability using the time evolution of radial distribution functions and bond lengths. Their framework assessed model stability across small molecules to solids in equilibrium MD systems. They highlighted the risk of relying solely on force accuracy for MLFF evaluation, offering new insights for the MLFF community. However, in semiconductor research, where phase transitions, interphase reactions, and defect dynamics are pivotal, systems often deviate from equilibrium. This points out the need for adapted metrics that are more aligned with semiconductor-specific material properties.

In addition, various condensed-phase datasets [31, 32, 33, 34, 35] have been shared in the existing literature and utilized for model evaluation. However, for the general ML community without prior experience in computational materials modeling, evaluating various material properties beyond energy and force using publicly available benchmarks is difficult. Thus, we offer not only an essential dataset for thin-film research, but also material property benchmarks for comprehensive model evaluation, allowing users with no prior experience in the field to evaluate models through actual simulations.

## 3 SAMD23 Datasets

SAMD23 includes two semiconductor material datasets: SiN and HfO. SiN is increasingly used in various semiconductor applications, particularly as a storage medium in advanced next-generation flash memory devices [36]. HfO is typically used as a high-k material and a crucial ferroelectric material in complementary metal-oxide-semiconductor technology, showing great potential for emerging electronics applications [37]. These two datasets are presented here to investigate the properties and behavior of SiN and HfO thin-film materials and provide a valuable resource for the development and evaluation of MLFF models. For details on dataset usage and generation methods, please refer to Section A.1.

**Composition.** The composition of the datasets is summarized in Table 1, and the details are listed in Tables A.1 and A.2. All the datasets were released in extended-xyz format (see Section A.5 for details). The SiN dataset encompasses various stoichiometric ratios, with atom counts ranging from 16 to 510. Meanwhile, the HfO dataset comprises 96 atoms, exhibiting a 1:2 ratio of 32 Hf atoms to 64 O atoms. More detailed information on characteristics of the datasets is provided in Section A.2.

### 3.1 Generation Methodology

To prepare the raw data, ab initio simulations were performed using Vienna ab initio simulation package (VASP) [38, 39, 40] with the Perdew-Burke-Ernzerhof generalized-gradient approximation exchange-correlation functional [41]. Convergence tests were conducted to determine the k-point meshes and cutoff energies for accurate DFT calculations, with a maximum energy convergence of 2 meV/atom and a maximum force error convergence of 50 meV/Å. The crystal structures of SiN and HfO were sourced from the Materials Project [42] and AFLOW databases [43] (see Section A.3 for details). For both SiN and HfO, we prepared ID and OOD datasets, which are exclusively sampled

from the raw data. The ID datasets were randomly divided into three groups of a ratio of 8:1:1, referred to as train, validation, and test sets, respectively.

**Silicon Nitride.** The SiN dataset was derived from 110 independent DFT simulations, encompassing diverse structures including amorphous and crystals ($\alpha$, $\beta$, and $\gamma$), one- or two-dimensional defects, surfaces, and isolated nano-clusters, which are illustrated in Figure A.1. Simulations were performed at various temperatures, strains, and ensembles, with the number of atoms per unit cell ranging from 16 to 510. The SiN simulations encompassed various stoichiometries of SiN as well as Si-only and N-only structures. The ID dataset consists of samples obtained from these simulations with a sampling interval of 9 fs. To generate the OOD dataset, we employed amorphous SiN as the initial structure, which was used to study the synthesizability of inorganic materials [44], and conducted a melt-quench-relaxation simulation. For additional details, please refer to Table A.2.

**Hafnium Oxide.** The HfO dataset was created using a modified melt-quench-annealing (m-MQA) method, in which temperature varied, as illustrated in Figure A.6. The MQA method, which inspired our m-MQA method, has been frequently used to create datasets [45, 46, 33]. To obtain high entropy structures, we incorporated three pre-melting stages at an extremely high temperature of 5000 K in the m-MQA method. For full method details, refer to Section A.4. We conducted 12 MD scenarios (yielding 60 simulation conditions) using the m-MQA method on five types of $HfO_2$ crystals (*i.e.*, monoclinic, tetragonal, cubic, and two orthorhombic structures) along with seven randomly generated structures. Each structure had 96 atoms (32 Hf and 64 O) to maintain the $HfO_2$ stoichiometry. Randomized structures were generated by randomly distributing 32 Hf and 64 O atoms within the unit cells of the $HfO_2$ crystals. Of the 12 simulations performed, simulations for 10 structures (five crystals and five random structures) and the remaining two random structures were utilized as the sources of the ID and OOD datasets, respectively. We sampled snapshots from each source with a sampling interval of 9 fs in the three pre-melting and melting stages and an interval of 12 fs in the quenching and annealing stages.

**SiN vs. HfO.** SiN data generation demonstrates a wider range of atomic configurations, encompassing various experimentally validated phases and defect structures. Conversely, HfO data generation employs the efficient m-MQA method, which is introduced in this report as a user-friendly approach for researchers proficient in DFT. Further, SiN data generation covers a wide range of stoichiometries and numbers of atoms (16-510 atoms), whereas HfO data generation focuses solely on a composition of Hf:O = 1:2, involving 96 atoms. Therefore, the model training results with the two datasets exhibited noticeably different aspects (see Section 4.2).

## 4 Numerical Benchmark: Energy and Force

MLFF models have their own training schemes on a target dataset, involving data preprocessing methods, loss functions, and hyperparameter settings, all of which can significantly influence the training results. To ensure a fair benchmark, we have empirically but carefully selected a common training scheme tailored for our datasets, as discussed in Section 4.1; the details are explained in Section C. Sections 4.2-4.4 provide the benchmark results on SiN and HfO, including an ablation study and discussions on the model inference speed and data scaling effect.

### 4.1 Benchmark Setting

**Error Metric.** Table C.1 lists the errors and loss functions used in previous MLFF studies. In this benchmark, we selected the sum of the root mean square errors (RMSEs) of the per-atom energy and force, denoted by the **EF metric**, as the metric for evaluating energy and force predictions for the following reasons.

For the energy error metric, two types of energy errors for a snapshot can be used: the total energy error and the per-atom energy error computed by dividing the total energy error by the number of atoms in a snapshot. However, the total energy error depends on the number of atoms in a snapshot, whereas the per-atom energy error remains invariant. Since our SiN dataset consists of snapshots of various sizes, we use the per-atom energy error. Subsequently, among the two per-atom energy errors, the mean absolute error (MAE) and RMSE, we employ **the RMSE of per-atom energy** as the energy error metric. For stable MD simulations, it is crucial to recognize the potential risks stemming from significant errors that can result in catastrophic failure. The RMSE can capture such risks when

assessing the extrapolation capability of MLFF models using OOD setsm rather than the MAE; this is evident in Section 4.2.

In the case of force error metrics, three error types are widely used in ML research: component-wise MAE, RMSE, and MAE with the size of force vectors (L2MAE). However, component-wise MAE has a critical flaw: the absence of rotational invariance (*i.e.*, if a snapshot is rotated, the component-wise absolute error of forces is changed), which is a highly recommended feature for a force field. Among the remaining two candidates, L2MAE and RMSE, which support rotational invariance, we employ **the RMSE of force** for the same reason as with the energy metric.

**Loss Function.** A loss function integrating the errors of energy and force is commonly used to train MLFF models. Since we employ the per-atom energy, the total loss function has the form (1),

$$\mathcal{L}_{tot} = \frac{\lambda_e}{|B|} \sum_{i \in B} \mathcal{L}_E \left( \frac{E_i}{n_i}, \frac{\hat{E}_i}{n_i} \right) + \frac{\lambda_f}{|B|} \sum_{i \in B} \mathcal{L}_{\mathbf{F}}(\mathbf{F}_i, \hat{\mathbf{F}}_i), \tag{1}$$

where $B$ indicates a batch of snapshots, $\mathcal{L}_E$ (or $\mathcal{L}_{\mathbf{F}}$) indicates energy (or force, respectively) loss, and $\lambda_e$ (or $\lambda_f$) is a weight of energy (or force, respectively) loss. In this benchmark, aligning the loss function to the EF metrics, we set $\mathcal{L}_E$ by the MSE of per-atom energy and $\mathcal{L}_{\mathbf{F}}$ by the MSE of forces; thus, we refer to it as **MSE-based loss**. Moreover, to secure generalization performance, we also tested setting $\mathcal{L}_E$ by the MAE of per-atom energy, and $\mathcal{L}_{\mathbf{F}}$ by the L2MAE of forces, adopted by some MLFF models that showed promising results [19, 47, 25]; the loss is called **MAE-based loss** in this paper. The exact formulations of the losses are shown in Table C.1. Generally, $\lambda_e$ is set by 1, and $\lambda_f$ are determined in various ways, such as manually setting it as a constant [16, 17, 19, 25], adjusting it based on the snapshot size [22, 23, 24], or gradually modifying it during the training [48]. In this benchmark, we set default $\lambda_f$ by 1, and performed an exploration study.

**Graph Generation.** To enable models to effectively capture atomic interactions, it is necessary to generate graphs from the snapshot, which represents point cloud data of atom coordinates. The most straightforward approach to convert snapshots to graphs is to build edges between nearby atoms using a cutoff radius; we set the cutoff radius to **6Å**. Further, owing to high training cost and insufficient GPU memory for some models, we restricted the maximum number of neighborhood atoms as **50**, from the nearest. The effect of the restriction is discussed in Section C.3.1.

## 4.2 Results and Analysis

We performed benchmarks using 10 models on our datasets: BPNN [13], DPA-1 [49], SchNet [16], DimeNet++ [18], GemNet-T/-dT [19], NequIP [22], Allegro [23], MACE [24], and SCN [25], all of which are described in Section B. The detailed training recipes are presented in Section C.2.

### 4.2.1 Model Comparison

Overall, as depicted in Figures 1 (a) and (b), the GNN-based models, except for SchNet, are superior to the two descriptor-based models, BPNN and DPA-1. Surprisingly, across all sets, DimeNet++ outperforms recently proposed GNN-based models. Nevertheless, as shown in Section 5.3, DimeNet++ exhibits a significantly low performance for one of the simulation indicators for HfO. This phenomenon is consistent with the results reported in a recent study [9] arguing that accurate force predictions do not fully represent the accuracy and applicability of MLFF models in simulations.

For every model, the EF metrics of the OOD set are larger than those of the test set, aligned with our intention. In both datasets, SchNet exhibits the largest discrepancy between the EF metrics of the train and test sets as well as the highest EF metric of the OOD set, indicating that SchNet has a potential risk of significant overfitting. BPNN, DPA-1, and Allegro also have relatively high EF metrics for the OOD set. Consistent with the results of the OOD set, these four models display unfavorable performance, as measured by the simulation indicators (see Section 5.3).

GNNs incorporating rotational equivariance (*e.g.*, NequIP, Allegro, and MACE) show relatively smaller differences between the EF metrics of the train and test sets compared with the other models, highlighting the advantageous aspects of rotational equivariance. However, in HfO, a substantial discrepancy between the EF metrics of the test and OOD sets is observed for Allegro and MACE. Considering that both models are based on the atomic cluster expansion (ACE) theory [28], searching for factors that contribute to such discrepancies would provide key insights for improving ACE-based GNNs.

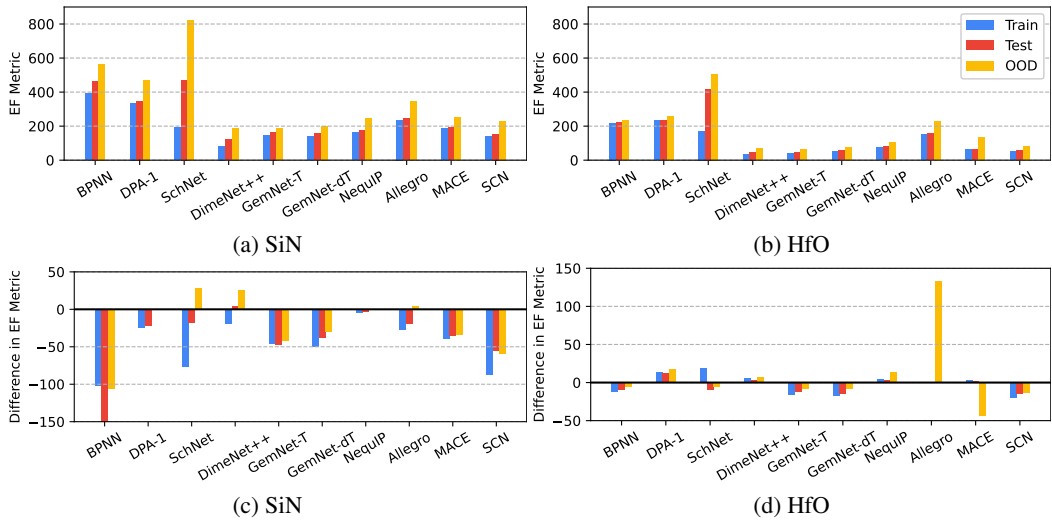

Figure 1: (a, b) EF metrics of the models trained with MSE-based loss. (c, d) Difference between EF metrics of models trained with MAE- and MSE-based losses, where the positive value infers training with MSE-based loss is superior.

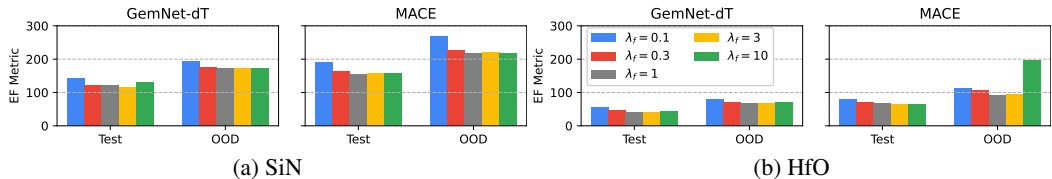

Figure 2: EF metrics obtained by models trained with varying $\lambda_f$ in training loss

### 4.2.2 Training Factor Exploration Study

**MSE- vs. MAE-based Losses.** Additionally, we trained the models using the MAE-based loss. Figures 1 (c) and (d) present an EF metric comparison between the models trained using the MAE-based loss and those trained using the MSE-based loss, respectively. The results obtained for SiN clearly indicate the substantial superiority of the MAE-based loss. However, for HfO, the superiority cannot be easily determined; in most cases, the differences of EF metrics are relatively small. Specifically, while Allegro shows a significant performance drop on the HfO OOD set, the accuracy of MACE is improved.

**Coefficient of Force Loss ($\lambda_f$).** For all models, force errors are dominant in the terms of EF metrics. Thus, if we design a training loss to reduce force loss by varying the force coefficient $\lambda_f$, the test force errors can be also reduced accordingly. To investigate the effect of $\lambda_f$, we trained GemNet-dT and MACE using the MAE-based loss with $\lambda_f$ ranged from 0.1 to 10; Figure 2 shows the results. Consistent with expectations, the models trained with $\lambda_f < 1$ show increased errors compared to $\lambda_f = 1$. However, the HfO results of MACE with $\lambda_f = 10$ reveal that increasing $\lambda_f$ does not always lead to a decrease in the force error and results with $\lambda_f > 10$ may be worse. Consequently, when exploring a proper $\lambda_f$ as a hyperparameter tuning, it is recommended to train MLFF models with $\lambda_f$ in the range of $1 \leq \lambda_f < 10$.

### 4.3 Model Exploration Study

As a goal of MLFF models, it is important to not only achieve the precision of DFT-based simulations, which provide reference data, but also make MD simulations as fast as possible. Thus, Figure 3 illustrates a Pareto plot between the inference time per snapshot and the EF metric of each model, where these two factors show a trade-off relationship. We performed the inference five times on the HfO test set with a batch size of 1, which mimics a step-by-step process in the simulations, using a V100 GPU. To further explore the tradeoff, we trained models whose feature dimensions varied from

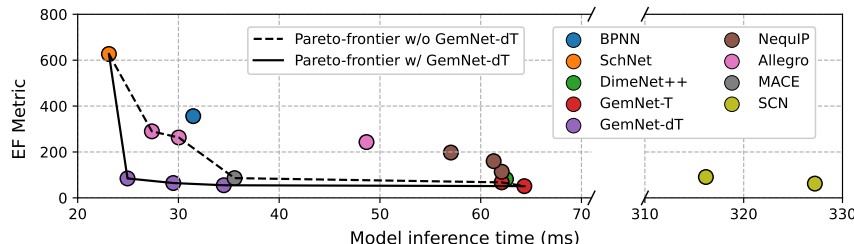

Figure 3: Pareto plot for inference time per snapshot vs. EF metric on HfO test set.

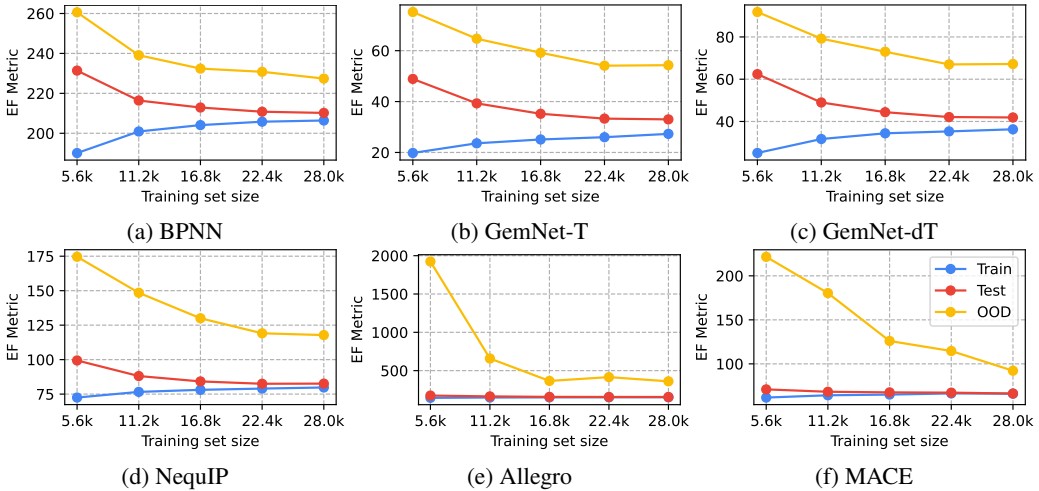

Figure 4: Data scaling effect. The EF metrics are obtained from six MLFF models trained by using 20%, 40%, 60%, 80%, and 100% snapshots of the HfO training set. For all the points of Test (red) and OOD (yellow) curves, the test and OOD sets are identically used to obtain EF metrics. At each point of Train (blue) curves, the correspondingly sampled training set is used.

the base architecture used in Section 4.2.1. The relative variation scale was selected from {0.25x, 0.5x, 2x}; the variation choices for each model are presented in Section C.4.2. To ensure clarity, no variant models within each model that fell outside the trade-off were included.

All GemNet-dT models are located on the Pareto-frontier curve. Similar to energy, GemNet-dT predicts forces only by the forward process in GNNs, instead of the conventional calculation in MLFFs, where the forces are calculated by the derivatives of energy with respect to the atom positions to satisfy the energy conservation law. The direct force prediction of GemNet-dT does not satisfy the energy conservation law, leading that the property benchmark results of GemNet-dT fall significantly short of those of GemNet-T (see Section 5.3). However, GemNet-T required a 2.2 times longer inference time and a much larger GPU memory than GemNet-dT, implying that GemNet-dT is more suitable for large-scale simulations by handling more atoms in less time. Therefore, when considering the aforementioned trade-off, a model development approach based on direct force prediction is a promising direction.

Although SchNet is the fastest, it yields the highest error, and thus, may generate unreliable simulations. SCN, which also adopts direct force prediction, achieves an EF metric comparable to that of GemNet-dT; however, its computational requirement is over 10 times higher, making it unsuitable for simulations. Except for GemNet-dT, Allegro and MACE show the Pareto-frontier results indicated by the dashed line. If the significant discrepancy in the EF metrics between the test and OOD sets is resolved, they can be employed in simulations where the energy conservation law should be satisfied.

## 4.4 Data Scaling Effect

We also present the data-scaling effect, which may be helpful in adopting a training strategy for MLFF researchers. We randomly sample 20%, 40%, 60%, and 80% snapshots from the HfO training

set (*i.e.*, 5.6k, 11.2k, 16.8k, and 22.4k), and trained models by 1000, 500, 334, and 250 epochs, respectively, to train models by the same iterations; the detailed setting is described in Section C.5 We selected six models and evaluated them on the HfO dataset.

As shown in Figure 4, for all models, as more training data are used, the training error increases while the test error decreases, and the gap between these two errors also decreases. In most cases, the OOD error decreases. These results suggest that training with fewer epochs and more data can improve the generalization performance of MLFF models. Meanwhile, the difference between the EF metrics of the models trained using 80% and 100% snapshots of the training set is marginal, indicating that performance cannot be improved using more data. This observation implies that the training set sampled from the raw dataset is sufficient for our semiconductor MLFF benchmark.

## 5 Property Benchmark: Simulation Indicators

To evaluate the accuracy and reliability of the models in predicting material properties, we introduce a set of five indicators: radial distribution function (RDF), angular distribution function (ADF), bulk modulus ($B_0$), equilibrium volume ($V_0$), and various potential energy curves (PECs). In this study, the indicators are categorized into two types depending on the necessity of the MD simulation during evaluation: RDF and ADF are classified as dynamic indicators, whereas the remaining indicators are classified as static indicators. An evaluation of any model using these indicators can be easily performed within our framework. By utilizing these indicators, researchers can enhance their understanding of the MLFF model behavior and performance.

### 5.1 Dynamic Indicators

The dynamic indicators, RDF and ADF, are derived from high-temperature MD trajectories, allowing us to assess the stability of the models in different atomic environments affected by turbulence or active movement induced by high thermal energy. For SiN (1200 K) and HfO (1200 K and 1800 K), high temperatures were carefully selected to investigate the model stability. On the one hand, RDF, also known as the pair correlation function, captures density changes relative to the distance from a chosen reference particle. On the other hand, ADF expands the analysis beyond radial distances by characterizing the angular distribution of particles surrounding a reference particle. The analysis of RDF and ADF is fundamental in simulations as it aids in identifying structures, phases, and interactions while providing a deeper understanding of the behaviors and reactions occurring within solids. We conducted evaluations using various structures and a wide range of supercells, consisting of 1,296-2,835 atoms for SiN and 2,592-3,456 atoms for HfO, highlighting the significant number of atoms involved (Table D.1). We have calculated representative values ($RDF^{ID}$, $RDF^{OOD}$, $ADF^{ID}$, and $ADF^{OOD}$) for the evaluation results of RDF and ADF; the details are described in Section D.1.

### 5.2 Static Indicators

The static indicators, $B_0$, $V_0$, and PECs, are derived from the relationships between input structure and energy, without requiring MD trajectories, offering valuable insights into the atomic and structural characteristics of materials.

**Bulk Modulus and Equilibrium Volume.** The Birch-Murnaghan equation of state (EoS) provides crucial information about thin-film materials, such as SiN and HfO, through two parameters: $B_0$ and $V_0$ [50]. These parameters are crucial for studying film stability, elasticity, and interfacial properties. $B_0$ signifies the response of films to pressure and external stress and indicates their flexibility, hardness, and resistance to volume changes. $V_0$ helps determine the films' thickness-residual stress relationship, enabling evaluation of their state (strained or relaxed) and refinement of the growth procedure. Understanding these parameters is vital for engineering thin-films with desired properties for potential applications in electronics, optoelectronics, and micro-electromechanical systems. To compare the models, we obtained the final scores for each model (*i.e.*, $B_0^{ID}$, $B_0^{OOD}$, $V_0^{ID}$, and $V_0^{OOD}$) through the EoS evaluation for different structures. For further information regarding the calculation, please refer to Section D.2.

**Potential Energy Curves.** The performance of the MLFF models at the atomic level was assessed using PECs. This indicator focuses on evaluating systems that have a sparse atomic environment with a small number of atoms, unlike the dense periodic boundary conditions encountered in our

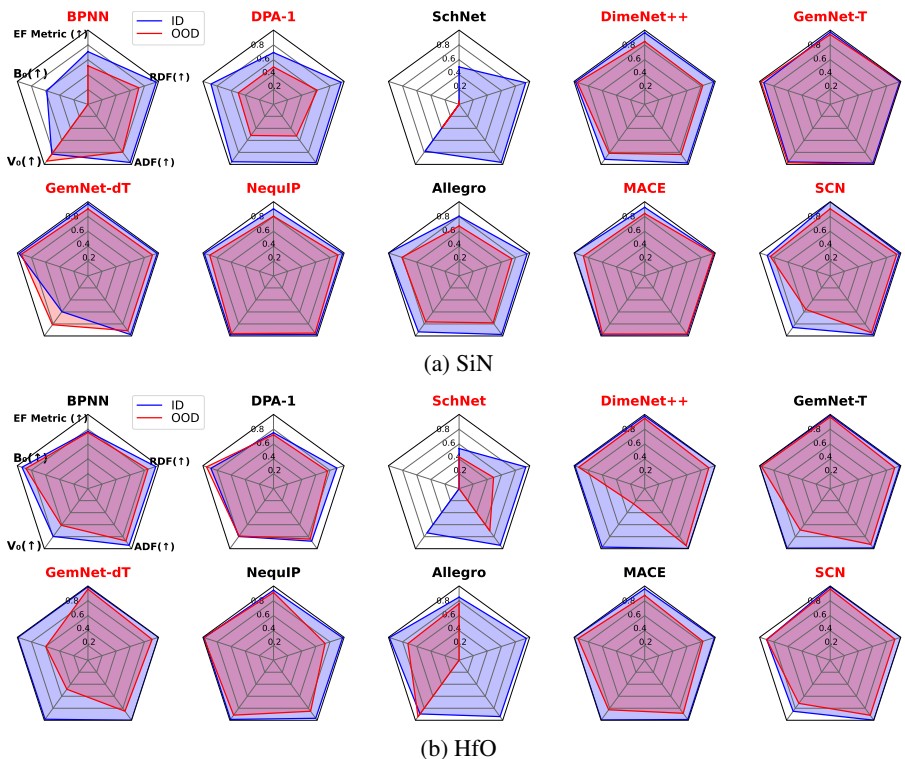

Figure 5: Comprehensive comparison of models based on EF metric and simulation indicators, where red and blue plots visualize the results of ID and OOD, respectively. Model names colored by black and red represent the models trained using MSE- and MAE-based loss, respectively.

solid-state structure dataset. By assessing the model's ability to accurately represent the interactions and dynamics in sparse environments, this evaluation serves as an OOD assessment. This is particularly relevant in scenarios involving molecular reactions, surface interactions, or nanoscale systems. Understanding the model's performance in these contexts provides valuable insights into its generalization capabilities beyond the bulk behavior observed in solid-state structures. For detailed information on PEC evaluations, please refer to Section D.3.

## 5.3 Results and Discussion

Figure 5 shows a comparison of the models based on the EF metric, incorporating dynamic and static indicators. For detailed information on the calculation of scores, please refer to Section D.4. Each model was trained with either MAE-based or MSE-based loss. Among them, the model with the highest sum of all indicator scores is shown in Figure 5.

**Model Comparison.** Overall, for both SiN and HfO, GemNet-T, NequIP, MACE, and SCN achieve prominent results; however for SiN, $V_0$ of SCN exhibits modest performance. The descriptor-based models, BPNN and DPA-1, show intermediate performance on both datasets. SchNet shows a significant mismatch in $B_0$ for both datasets and have the poorest generalization performance.

**Correlations of Metrics.** In the case of HfO, DimeNet++ and GemNet-dT achieve excellent results in terms of their EF metric scores for the ID and OOD sets, as shown in Figure 1. However, the performances of the two models in terms of the simulation indicators for OOD are not consistent. DimeNet++ shows a significant decrease in OOD performance for $V_0$ prediction, while GemNet-dT exhibits a substantial decrease for both $V_0$ and $B_0$ predictions. This indicates the limitations of the existing evaluation approach, which is primarily focused on energy and force, emphasizing the need for evaluation metrics related to material simulations and properties. For SiN, DimeNet++ and GemNet-dT do not show such a pronounced trend, which could be attributed to the characteristics of the dataset.

Unlike SiN, HfO was generated using the m-MQA method at only one composition ratio (1:2), thus making it relatively challenging to evaluate OOD performance compared to SiN, which was created by considering various defects, compositions, structures, and environments. Although evaluating the EF metric for OOD alone cannot guarantee the performance of the simulation indicators, the EF metric for OOD exhibited a higher Pearson correlation with the simulation indicators than the EF ID evaluation (see Figure D.7). This suggests that the simulation indicators can serve as a guide for evaluating the extrapolation performance of MLFF models.

**Simulation Failure.**   In terms of the OOD evaluation of the dynamic indicators, Allegro performs moderately on SiN but scores 0 on HfO. This difference is due to the interrupted HfO simulations caused by unrealistic atomic collisions during high-temperature simulations (Figure D.8). These interrupted simulations show abnormal atomic movements, with two Hf atoms attracting each other despite being too close, violating the expected repulsion behavior and terminating the simulation. These issues in HfO can be attributed to the differences in the datasets. In the SiN dataset, simulations of single element systems, such as dense structures consisting solely of Si and N elements, were included, whereas the HfO dataset lacked such simulations. Consequently, relatively more datapoints in the SiN training set encompassed situations with short interatomic distances, allowing the model to better learn and handle these scenarios. The variation in dataset characteristics is also evident in the PEC evaluation, which is discussed in further detail in Section D.3.2.

## 6   Conclusion

We introduced an MLFF benchmark suite consisting of SAMD23 datasets with two semiconductor materials and ten state-of-the-art models with six evaluation metrics. The experimental results show that benchmarking only the energy and force prediction errors is insufficient, and simulation-based metrics should be used to reflect the practical utility of a model. In addition, evaluation using the OOD sets is important to correctly understand the prediction performance of models. Moreover, although recently developed GNN-based models such as GemNet-T, NequIP, and MACE have shown relatively better simulation quality, no clearly superior models emerged. If we consider the predictive performance of energy and forces and inference time for large-scale simulations, GemNet-dT becomes a viable option; however, it should be improved to operate properly in simulations. We anticipate that our benchmarks will encourage future developments in the field, resulting in MLFF models that satisfy the accuracy, speed, and scalability requirements. We hope that our benchmark will provide valuable insights into the field of electronics.

**Limitations.**   The model that was shown to perform well in other fields (*e.g.*, molecules) may have been underestimated in this benchmark. Unlike datasets focused on small molecule structures, this study considered condensed-phase systems containing many atoms, making it practically impossible to include all possible atomic environments in the dataset. In addition, simulations handling long-range interactions beyond a few hundred nanometers were not included in the training data, because the high computational cost of ab initio simulations limits the number of atoms. Despite using periodic boundary conditions and including larger structures, our DFT-based data generation scheme cannot fully account for long-range interactions.

**Future Plans.**   In future works, we intend to continuously incorporate state-of-the-art MLFF models and expand and share datasets with various compositions openly. Specifically, in the immediate future, we will introduce datasets on the Hf-Si-O and Hf-Zr-O systems, incorporating diverse chemical stoichiometries to ensure their broad applicability across a wide spectrum of compositions. In addition, plans are in motion to sequentially release diverse semiconductor datasets, including ternary systems or higher, with constituents such as Si, Ge, C, N, O, H, Hf, Zr, B, P, Ga, and As. This extension will guarantee the exploration of various atomic ratios, allowing for a more comprehensive evaluation of model performance. As training models from scratch for every appending dataset is inefficient, we are planning to adopt efficient training schemes such as transfer learning, which we expect to reduce the cost of model training for growing datasets. It is also important to subjoin additional metrics and loss factors such as stress to evaluate the compatibility with various simulation conditions (*e.g.*, NPT), and to reflect indicators during training to provide better prediction performance in practice. Finally, since the main purpose of MLFFs with respect to semiconductors is to enable large-scale simulations that traditional ab initio methods cannot perform, we will extend our benchmark to include the compression and parallelization performance of the models.

## Acknowledgments

We express our gratitude to Albert Musaelian, the lead researcher of the NequIP and Allegro in Materials Intelligence Research group at Harvard University, for providing us with an opportunity to enhance the fairness of our benchmark results. In January 2024, he informed us that an updated configuration of Allegro succeeded in achieving a lower EF metric, surpassing the performance of the configuration we utilized in our study before February 2023. Our forthcoming arXiv manuscript will present additional results based on this updated Allegro configuration.

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

# Appendix

## A  Details of Datasets

Table A.1: Overview of total simulations performed for the HfO dataset. $N_{cond}$ represents the total number of simulation conditions, $N_{str}$ is the total number of MD snapshots, and $N_f$ indicates the total number of atomic force data. Structures with index (idx.) indicate the same crystal family but have different lattice parameters.

| Element | Structure (idx.) | Simulation Condition | $N_{cond}$ | $N_{str}$ | $N_f$ |
|---|---|---|---|---|---|
| Hf & O | Monoclinic | m-MQA method | 6 | 16,000 | 1,536,000 |
| | Tetragonal | m-MQA method | 6 | 16,000 | 1,536,000 |
| | Cubic | m-MQA method | 6 | 16,000 | 1,536,000 |
| | Orthorhombic (1) | m-MQA method | 6 | 16,000 | 1,536,000 |
| | Orthorhombic (2) | m-MQA method | 6 | 16,000 | 1,536,000 |
| | Randomized Structures (ID) | m-MQA method | 30 | 80,000 | 7,680,000 |
| Hf & O | Randomized Structures (OOD) | m-MQA method | 12 | 32,000 | 3,072,000 |

Table A.2: Overview of total simulations performed for the SiN dataset. $N_{cond}$ represents the total number of simulation conditions, $N_{str}$ is the total number of MD snapshots, and $N_f$ indicates the total number of atomic force data.

| Element | Structure | Simulation Condition | $N_{cond}$ | $N_{str}$ | $N_f$ |
|---|---|---|---|---|---|
| Si & N | Amorphous | High T MD (2000, 4000 K; w/o strain) | 11 | 13,650 | 1,006,500 |
| | | High T MD at 1500K (w/ strain ±7%) | 4 | 4,000 | 306,000 |
| | | Quenching (4000 to 300 K; w/o strain) | 8 | 26,150 | 1,305,600 |
| | | Quenching (4000 to 300 K; w/ strain ±7%) | 4 | 2,000 | 153,000 |
| | | Structure Relaxation (w/o strain) | 11 | 1,274 | 77,952 |
| | | Structure Relaxation (w/ strain ±7%) | 4 | 570 | 38,760 |
| | Crystals ($\alpha, \beta, \gamma$) | MD (300, 1200, 2100 K) | 9 | 18,000 | 504,000 |
| | | Structure Relaxation | 9 | 569 | 15,932 |
| | Defect Structures | MD with Divacancy (2000 K) | 3 | 300 | 87,600 |
| | | MD with Grain-Boundary (2000 K) | 5 | 500 | 103,600 |
| | Surfaces | MD (300, 1200 K) | 6 | 6,000 | 480,000 |
| | | Structure Relaxation | 6 | 1,400 | 105,600 |
| | Isolated Clusters | MD (300, 1200 K) | 6 | 1,200 | 142,400 |
| | | Structure Relaxation | 6 | 600 | 70,800 |
| Si | Crystals (Diamond, SC, BCC, FCC) | MD (2100 K; w/o strain) | 4 | 2,000 | 69,500 |
| | | MD (2100 K; w/ strain ±7%) | 8 | 4,000 | 139,000 |
| | Defect Structures | MD with Divacancy (2100 K) | 1 | 150 | 32,100 |
| | | MD with Two Vacancies (2100 K) | 1 | 100 | 51,000 |
| N | $N_2$ Molecules | MD (2100, 4000 K) | 4 | 2,000 | 128,000 |
| Si & N | Amorphous (OOD) | Melt, Quench, and Relaxation | 3 | 3,700 | 388,500 |

### A.1  Guidance

Simulation data can be flexibly utilized by researchers to study interatomic potential models according to their specific preferences. For example, the HfO dataset was generated using the m-MQA method, which involved initial structures encompassing various crystals as well as randomized structures. In our HfO dataset, we included both types of structures to provide researchers with a comprehensive dataset. Researchers have the flexibility to train their models using a subset of the dataset consisting of randomized structures and evaluate their performance on the remaining portion of the dataset comprising crystal structures, and vice versa. This approach enables thorough testing and assessment of model performance across different structural configurations within the HfO dataset.

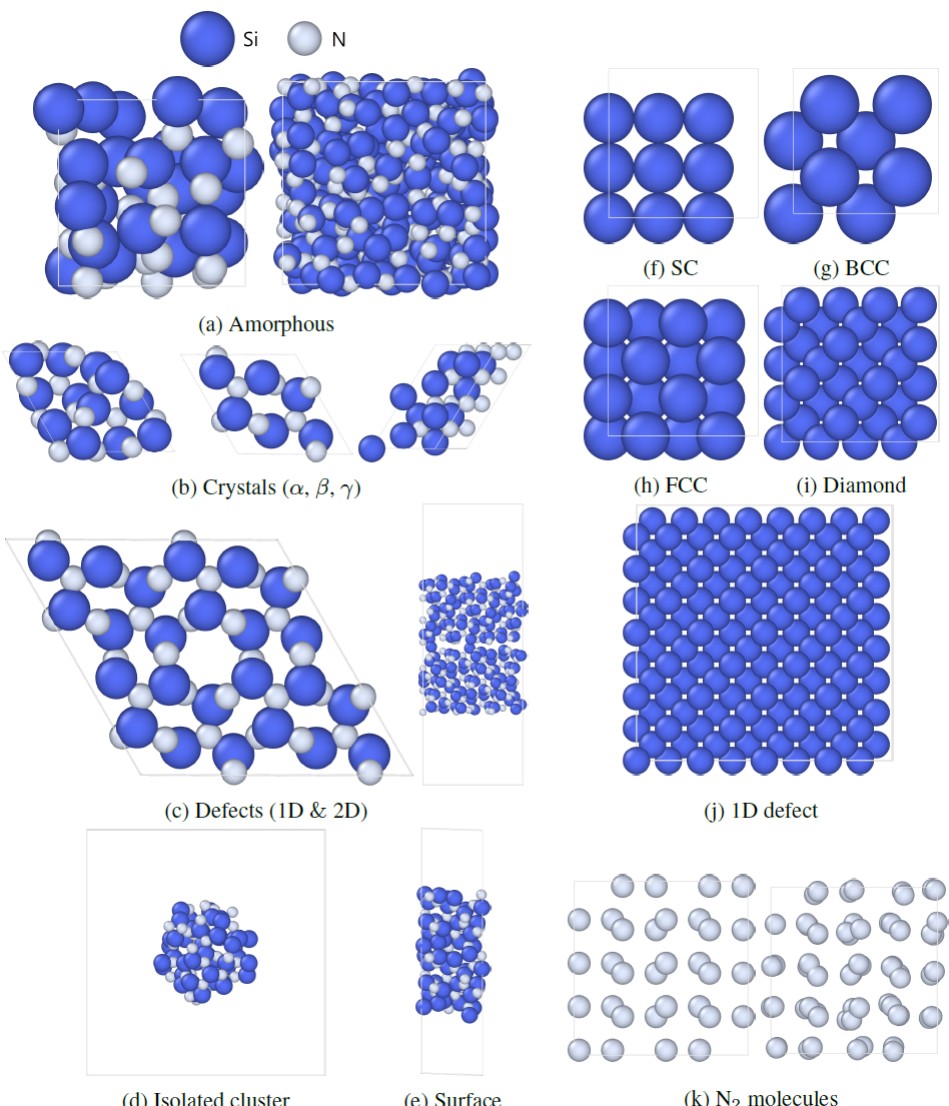

Si N

(a) Amorphous

(b) Crystals ($\alpha$, $\beta$, $\gamma$)

(c) Defects (1D & 2D)

(d) Isolated cluster

(e) Surface

(f) SC

(g) BCC

(h) FCC

(i) Diamond

(j) 1D defect

(k) $N_2$ molecules

Figure A.1: Reference atomic structures employed in the SiN dataset.

Researchers can modify the SiN simulation data to investigate the generalization performance of MLFF models. Despite the integrated SiN dataset containing not only SiN compounds but also Si-only structures and N-only structures, researchers have the choice to exclusively use SiN compounds for model training. Subsequently, they can evaluate the generalization performance of these models on Si- and N-only structures. However, it is crucial to recognize the significant challenge arising from the substantial disparities in sample distributions between SiN compounds and the Si- and N-only structures.

The generation of specialized datasets for silicon nitride necessitates a high degree of domain expertise. However, our proposed m-MQA scheme represents a notable advancement by presenting a streamlined approach for generating the HfO dataset. In contrast to conventional methods reliant on specialized knowledge, this scheme empowers researchers familiar with DFT to effortlessly generate the dataset. By simplifying the dataset generation process and relieving the requirement for extensive domain expertise, our m-MQA scheme enhances accessibility for researchers, facilitating advancements in MLFF research.

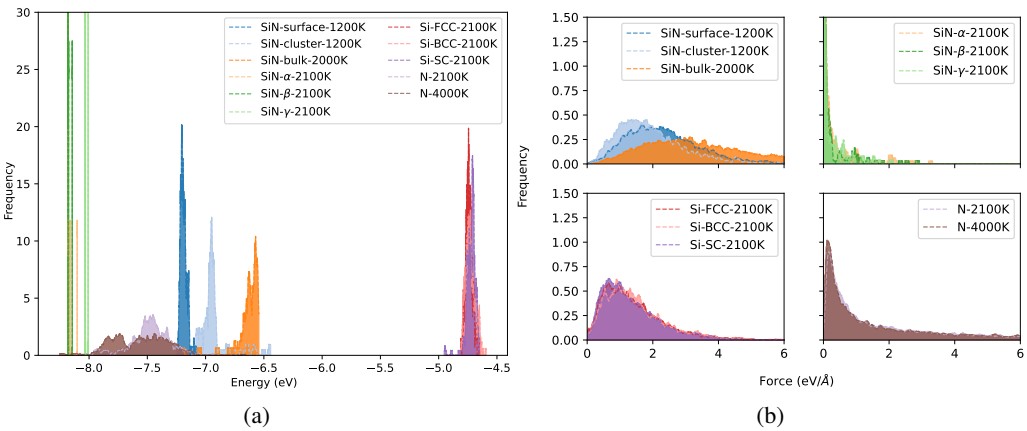

Figure A.2: Distribution of energy per atom and forces for SiN datasets: (a) energy per atom and (b) force per atom.

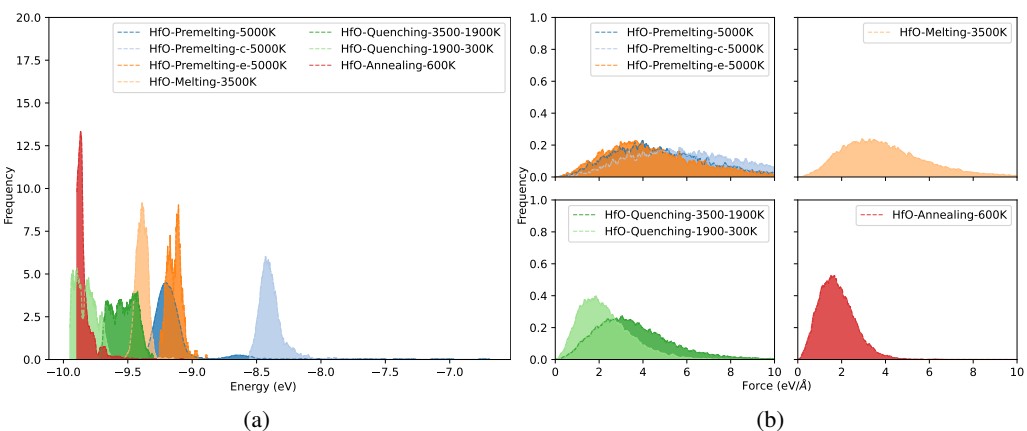

Figure A.3: Distribution of energy per atom and forces for selected MD simulations for HfO: (a) energy per atom and (b) force per atom.

## A.2 Dataset Characteristics

To illustrate the diversity of datasets of our DFT calculations, we analyze the energy and force distribution for each DFT calculation. Figure A.2 shows the energy and force distributions of the representative crystal structures in the SiN datasets. For brevity, we only plot the energy and force distribution of single MD scenario for each structure. The SiN dataset contains numerous phases, showing well-separated variations for both energy and forces. The force distribution is particularly interesting since it reflects the structural properties of each MD simulation. The force distribution shows a considerable variation ranging from 0 to 6 eV/Å for amorphous phases. However, the highly symmetric crystalline phases ($\alpha$, $\beta$, and $\gamma$-SiN) show that most of the forces on atoms are zero. MD simulations on face-centered cubic (FCC), body-centered cubic (BCC), and simple cubic (SC) also show relatively large variations of forces, but their distributions are narrower than amorphous cases. The N-only cases show similar distribution as the SiN-crystalline cases.

Unlike SiN, the HfO datasets consist of 96 atoms with a fixed Hf:O stoichiometry of 1:2. Each MD scenario shows variations in energy and force, as demonstrated in Figure A.3 (a). The high-temperature pre-melting stage at 5000 K shows the largest energy distribution. Depending on the compressive (HfO-premelting-c-5000K) or expansive (HfO-premelting-e-5000K) strain, they exhibit well-separated energy distribution. The average energy also lowers as the temperature decreases through the melting and quenching process. For the annealing process, they show the narrowest energy distribution. The force distributions of these MD scenarios also follow the typical patterns shown in MD simulations. For liquid-like phases that can be observed in pre-melting and melting

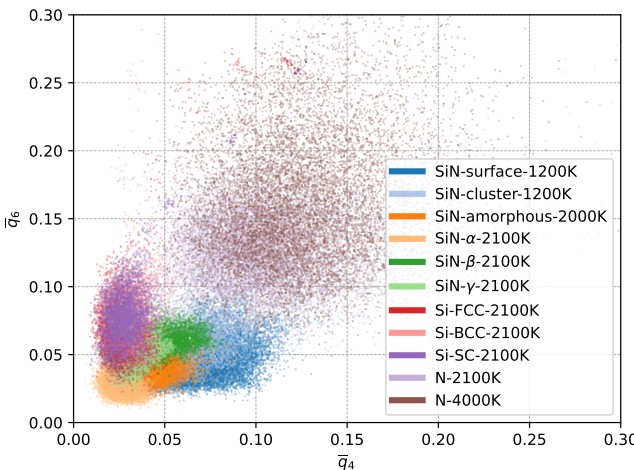

Figure A.4: Distribution of the averaged bond order parameter $\bar{q}_4$ and $\bar{q}_6$ for SiN.

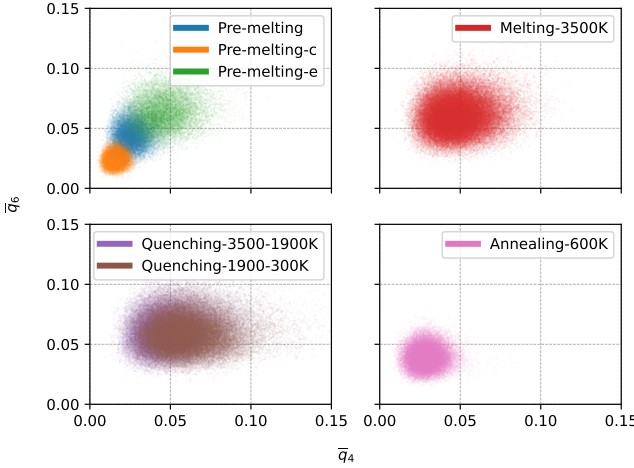

Figure A.5: Distribution of the averaged bond order parameter $\bar{q}_4$ and $\bar{q}_6$ for HfO.

stages (Figure A.3 (b)), we see relatively broader force distributions. As the temperature decreases through the quenching and annealing process, the average force on each atom decreases.

Another essential aspect of the MLFF dataset is the inclusion of diverse atomic environments. Since the local atomic environment determines the atomic energy and forces, having a diverse atomic environment in the dataset is crucial. One way of quantifying the local environment is the bond order parameter (BOA) [51]. We employ the averaged BOA, usually denoted by two-dimension vectors $\bar{q}_4$ and $\bar{q}_6$ [52]. This parameter can be a simple but good indicator for quantifying the diversity of the local atomic environment.

Figures A.4 and A.5 show the averaged BOA of SiN and HfO, respectively. SiN shows distinct phase separation in the $\bar{q}_4 - \bar{q}_6$ plane, with most BOA values falling between $0 \leq \bar{q}_4 \leq 0.3$ and $0 \leq \bar{q}_6 \leq 0.3$. HfO also exhibits similar phase separation, with unique patterns for each MD scenario. For the high-temperature pre-melting stage, the phase boundary is separated by a change in volume. For the melting and quenching stages, their coverage overlaps. However, the energy and force distribution among these MD scenarios are different as shown in Figure A.3. This means that the local atomic environment also differs even though $\bar{q}_4$ and $\bar{q}_6$ shows similar distribution. It is important to acknowledge that the BOA does not take into account atomic species and distance information, which can limit its reliability as a comprehensive descriptor. However, this analysis can provide an initial insight into the dataset's quality, serving as a starting point for further investigations.

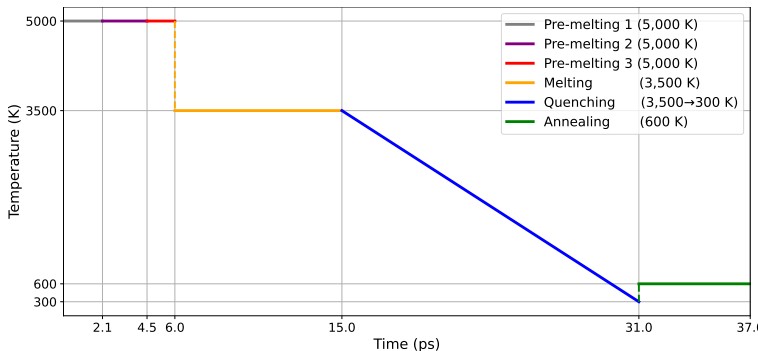

Figure A.6: A modified Melting-Quenching-Annealing scheme used for generating HfO dataset.

### A.3 Utilization of Large-Scale Structural Databases

The Materials Project [42] and AFLOW [43] are comprehensive databases that encompass a wide range of atomic structures, spanning various elements and structural motifs. These structures are predominantly derived from precise quantum mechanical calculations and represent materials in their energetically stable states, where the forces on most atoms approach zero.

For those diving into MD simulations via MLFF models, understanding the nature of these databases is crucial. They primarily present configurations where atoms are at their stable states with minimum-energy. The MD simulations, on the other hand, often explore high-energy conditions and a variety of force environments. Such scenarios are underrepresented in these databases, and thus the databases are less suitable for training MLFF models directly. However, their extensive collection of stable crystal structures can be initial configurations when curating or generating an MLFF training dataset.

### A.4 Details of the Modified Melt-Quench-Annealing Method

The generation of the HfO dataset involved the use of a modified melt-quench-annealing (m-MQA) method, where the temperature was scheduled as depicted in Figure A.6. In the m-MQA method, aimed at obtaining high entropy structures, we incorporated three pre-melting stages at an exceedingly high temperature of 5000 K. These stages involved pre-melting at crystal's original volume (pre-melting 1), followed by 10% isotropic compression (pre-melting 2) and 10% expansion (pre-melting 3) relative to the original volume. These stages were taken to ensure the presence of both dense and sparse atomic environments in the resulting dataset. Initial structures underwent three pre-melting stages at 5000 K for 6 ps, melting at 3500 K for 9 ps, quenching from 3500 to 300 K for 16 ps. Finally, the structures were restored to their original volume and annealed at 600 K for 6 ps. All simulations were performed under the NVT ensemble with the Nose-Hoover thermostat.

### A.5 Dataset Instruction

Our datasets can be downloaded when following the instructions in the our github url (https://github.com/SAITPublic/MLFF-Framework). There are four datasets of the extended-xyz format: raw and sampled datasets of SiN, and raw and sampled datasets of HfO. The sampled datasets are a portion of the raw datasets, and used in the experiments of this benchmark.

The extended-xyz format begins with a line declaring the total number of atoms. The subsequent line typically encompasses lattice information, energy, and other relevant details associated with the atomic structure for that specific simulation step. From the third line, each line represents an individual atom, where the first column designates the atomic type, followed by its Cartesian coordinates and force components. This structured format is interspersed throughout the file, capturing snapshots from various points in the simulation, providing a comprehensive record that offers insights into the dynamic behavior of atoms at different stages.

While the extended-xyz format can be handled with conventional text-processing utilities, employing specialized computational toolkits considerably streamlines the process. The Atomic Simulation Environment (ASE) [53], renowned for its versatility, effortlessly accommodates a wide range of

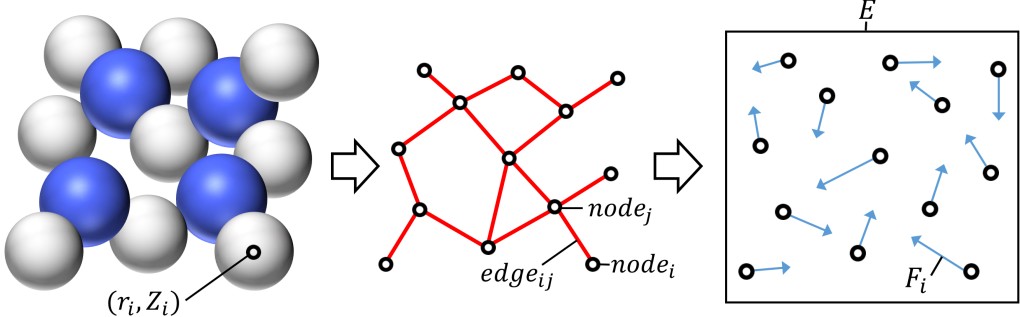

| (a) [Input] An atomic structure | (b) [Preprocessed] Graph (Optional) | (c) [Output] Energy and Forces |

Figure B.1: General flow of MLFF models.

atomistic formats, including the extended-xyz. From intuitive data visualization to comprehensive atomic analysis, its functionality significantly enhances user efficiency. Similarly, the Python Materials Genomics (pymatgen) [54] presents itself as an invaluable tool for researchers focused on atomic structural analysis. It offers deep insights into atomic structural data, enabling seamless structural analysis and transformations. Both ASE and pymatgen serve as commendable aids, optimizing the user experience in managing atomic structural databases.

## B  Details of Benchmark Models

In this Section, we briefly review the MLFF models used in this benchmark. Figure B.1 describes general flow of MLFF models for a snapshot. Figure B.1a illustrates a structure of input data, which is an 3D point cloud data consists of coordinates $r_i = (x_i, y_i, z_i) \in \mathbb{R}^3$, and atom numbers $Z_i \in \mathbb{Z}$ for $1 \leq i \leq n$ where $n$ represents the number of atoms. Figure B.1b describes the preprocessed data with 3D graph form, by constructing edges between points within a cutoff radius. As shown in Figure B.1c, model output consists of total system energy $E \in \mathbb{R}$, which is the sum of atomic features, and force $F_i = (F_i^x, F_i^y, F_i^z) \in \mathbb{R}^3$ for all atoms in the snapshot.

**BPNN [13]**  As a pioneer study for MLFF, BPNN introduced a base idea for the MLFF field: the total energy is represented by a sum of atomic features. For a regression model, a neural network consisting of fully-connected layers is employed. To train the neural network, the coordinates of atoms are converted into hand-crafted features by using symmetry functions to describe local environment of atoms; the hand-crafted features are generally referred to as descriptors.

**DPA-1 [49]**  DPA-1, which was specifically designed to be operated on DeePMD-kit [48], is an end-to-end deep potential energy model that includes trainable descriptors. These descriptors are invariant under translation, rotation, and permutation. It also employs a self-attention mechanism to effectively incorporate angular information for improved performance.

**SchNet [16]**  SchNet is a deep neural network to adaptively learn atomic features representing local atomic environment from atom-centered symmetry functions without relying on pre-defined descriptors. It combines atomic intermediate features using continuous-filter convolutions to mimic atomic interactions.

**DimeNet++ [18]**  DimeNet [17] is an MLFF model based on message passing GNNs, where directional information of atomic environment is embedded into directional messages. It utilizes distance features and dihedral angle features based on radial basis function and spherical basis function, respectively. As an improved version of DimeNet, DimeNet++ introduce an efficient implementation of the directional message passing layer of DimeNet, and its neural architecture is modified from that of DimeNet, enhancing predictive capabilities.

**GemNet (GemNet-T and -dT) [19]**  GemNet is a GNN-based model that incorporates directed edge embeddings and two-hop message passing, allowing the model to capture complex directional

information. Quadruplet, a tuple with four atoms, can be used for these scheme; GemNet using the quadruplet is called GemNet-Q. In GemNet-T, the message and intermediate features are described by using atom pairs and triplets, which is a tuple with three atoms. As an alternative to calculate forces, a direct force prediction method is proposed and applied into GemNet-T architecture, which is GemNet-dT. GemNet achieves universal approximation capabilities for translation-invariant and permutation/rotation-equivariant predictions.

**NequIP [22]**    NequIP utilizes E(3)-equivariant convolutions to capture interactions of geometric tensors, including vectors and higher-order tensors, of atom embeddings. NequIP can learn a representation of atomic environments more comprehensively, compared to models that use invariant convolutions and operate solely on scalars. Hence, it yields prominent accuracy on a diverse range of molecules and materials while requiring significantly fewer training data.

**Allegro [23]**    In GNN-based MLFF models, message passing is considered as a necessary technique to learn many-body interactions. However, the message passing needs information exchange among atoms, leading a large communication overhead on distributed computing system, which is unsuitable for large-scale simulation. To address this issue, Allegro combines equivariant features and tensor products to represent a strictly local equivariant atomic features without relying on message passing, achieving both accuracy and scalability.

**MACE [24]**    MACE is theoretically based on multi-ACE framework [55] that was extended from atomic cluster expansion (ACE) framework by introducing GNNs equipped with high body-order messages. Due to the usage of high body-order messages, MACE can reduce message passing layers to learn the atomic potentials, enabling fast and parallelizable computations and enhancing the training efficiency.

**SCN [25]**    SCN is a GNN-based model where atom features are a set of spherical channels represented by spherical harmonics, enabling it satisfy rotation equivariance. By relaxing the constraint of rotational equivariance in message passing and aggregation, SCN demonstrated that the performance of energy and force prediction can be improved. SCN is specifically designed for OC20 and requires heavy computation and memory usage. To deal with this issue, we reduced its model size by using 4 interaction (message passing) layers and 64 spherical channels; the small SCN was used for the all of our experiments.

## C    Details of Numerical Benchmark

### C.1    Error Metrics and Losses

Errors of energy and forces are commonly employed as evaluation metrics to assess the predictive performance of MLFF models, primarily due to their ease of use and straightforward interpretability. The errors and loss functions can be computed in several ways, which are listed in Table C.1. Among the metrics, as in discussed in Section 4.1, we chose **the RMSE of per-atom energy**, which can be considered invariant among various snapshot sizes, and **the component-wise RMSE of forces** (briefly called the RMSE of forces). In addition, we also express the **MSE-based loss** and **MAE-based loss**.

### C.2    Training Hyperparameters

In the ML community, where the reproducibility of experiments is highly valued, sharing training recipes that include hyperparameters for training MLFF models is considered a valuable endeavor. In alignment with this attitude, we present the hyperparameters of the models, which we trained for this benchmark study. The configuration files can be also accessed in our open-sourced code. By doing so, we aim to promote transparency and enable others to replicate and build upon our work in a meaningful way.

The all of configurations including neural architecture hyperparameters and optimization hyperparameters are prepared into our codes in Supplementary materials: codes.zip (in configs/train/SiN/ and configs/train/HfO/). Also, the training factors are listed in Table C.2.

Table C.1: Error metrics and losses used in MLFF research and this benchmark. $N$ is the number of snapshots, $n_i$ is the number of atoms in snapshot with index $i$. The normal characters such as $E_i$, $f_x^{i,k}$ indicate reference data (*i.e.*, ground truth). The characters with a hat such as $\hat{E}_i$, $\hat{f}_x^{i,k}$ indicate predicted values by MLFF models. $\lambda_f$ is a coefficient of force loss.

| Type | Name | Formulation |
|---|---|---|
| Energy error metric | MAE of (total) energy | $\sum_{i=0}^{N} |E_i - \hat{E}_i| / N$ |
| | RMSE of (total) energy | $\left[ \sum_{i=0}^{N} |E_i - \hat{E}_i|^2 / N \right]^{1/2}$ |
| | MAE of per-atom energy | $\sum_{i=0}^{N} \left| \frac{E_i - \hat{E}_i}{n_i} \right| / N$ |
| | RMSE of per-atom energy | $\left[ \sum_{i=0}^{N} \left| \frac{E_i - \hat{E}_i}{n_i} \right|^2 / N \right]^{1/2}$ |
| Energy loss | MAE of per-atom energy | $\left| \frac{E_i - \hat{E}_i}{n_i} \right|$ |
| | MSE of per-atom energy | $\left| \frac{E_i - \hat{E}_i}{n_i} \right|^2$ |
| Force error metric | Axis-wise MAE of force | $\sum_{i=0}^{N} \sum_{k=0}^{n_i} \left( |f_x^{i,k} - \hat{f}_x^{i,k}| + |f_y^{i,k} - \hat{f}_y^{i,k}| + |f_z^{i,k} - \hat{f}_z^{i,k}| \right) / \sum_{i=0}^{N} 3n_i$ |
| | Axis-wise RMSE of force | $\left[ \sum_{i=0}^{N} \sum_{k=0}^{n_i} \left( |f_x^{i,k} - \hat{f}_x^{i,k}|^2 + |f_y^{i,k} - \hat{f}_y^{i,k}|^2 + |f_z^{i,k} - \hat{f}_z^{i,k}|^2 \right) / \sum_{i=0}^{N} 3n_i \right]^{1/2}$ |
| Force loss | Axis-wise MAE of force | $\sum_{k=0}^{n_i} \left( |f_x^{i,k} - \hat{f}_x^{i,k}| + |f_y^{i,k} - \hat{f}_y^{i,k}| + |f_z^{i,k} - \hat{f}_z^{i,k}| \right) / 3n_i$ |
| | Axis-wise MSE of force | $\sum_{k=0}^{n_i} \left( |f_x^{i,k} - \hat{f}_x^{i,k}|^2 + |f_y^{i,k} - \hat{f}_y^{i,k}|^2 + |f_z^{i,k} - \hat{f}_z^{i,k}|^2 \right) / 3n_i$ |
| | L2-MAE of force | $\sum_{i=0}^{n_i} \left( |f_x^{i,k} - \hat{f}_x^{i,k}|^2 + |f_y^{i,k} - \hat{f}_y^{i,k}|^2 + |f_z^{i,k} - \hat{f}_z^{i,k}|^2 \right)^{1/2} / n_i$ |
| | L2-MSE of force | $\sum_{i=0}^{n_i} \left( |f_x^{i,k} - \hat{f}_x^{i,k}|^2 + |f_y^{i,k} - \hat{f}_y^{i,k}|^2 + |f_z^{i,k} - \hat{f}_z^{i,k}|^2 \right) / n_i$ |

**EF metric** = RMSE of per-atom energy + Axis-wise RMSE of force

**MAE-based loss** = MAE of per-atom energy + $\lambda_f \times$ L2-MAE of force
**MSE-based loss** = MSE of per-atom energy + $\lambda_f \times$ Axis-wise MSE of force

Table C.2: Details of training factors for 10 MLFF models.

| Model | Optimizer | Initial learning rate | Batch size SiN | HfO | etc. |
|---|---|---|---|---|---|
| BPNN | Adam | 5.e-3 | 16 | 16 | weight decay: 1.e-6 |
| DPA-1 | Adam | 1.e-3 | 16 | 16 | |
| SchNet | Adam | 1.e-4 | 16 | 16 | |
| DimeNet++ | Adam | 1.e-4 | 3 | 8 | |
| GemNet-T | AdamW | 5.e-4 | 4 | 8 | AMSGrad, ema decay: 0.999, gradient clipping: 10 |
| GemNet-dT | AdamW | 5.e-4 | 4 | 8 | AMSGrad, ema decay: 0.999, gradient clipping: 10 |
| NequIP | Adam | 5.e-3 | 16 | 16 | ema decay: 0.99 |
| Allegro | Adam | 5.e-3 | 16 | 16 | ema decay: 0.99 |
| MACE | Adam | 1.e-2 | 16 | 16 | weight decay: 5.e-7, ema decay: 0.99 |
| SCN | AdamW | 4.e-4 | 6 | 8 | AMSGrad, ema decay: 0.999 |

As a variant of Adam [56], AMSGrad [57] maintains the maximum of the past squared gradients instead of using exponential moving average, leading to more stable convergence. However, we empirically observed that training losses of MLFF models are often skyrocketed, resulting in relatively slow convergence. Thus, for models that used AMSGrad in the original papers, we tried to train models with or without AMSGrad, and then the better results are reported in this paper. Recently-proposed models tend to employ exponential moving average (EMA), which may also help to obtain stable training results.

While most of the hyperparameters of each model including the neural architecture introduced in the original paper (except SCN) were preserved, we established a common rule of some hyperparameters which can affect the training cost: epochs, batch size, and learning rate schedule. For both HfO and SiN datasets, MLFF models were trained during 200 epochs. The training batch size for each model was selected from the options of 3, 4, 8, 16 to ensure compatibility with the memory capacity of the V100 GPU.

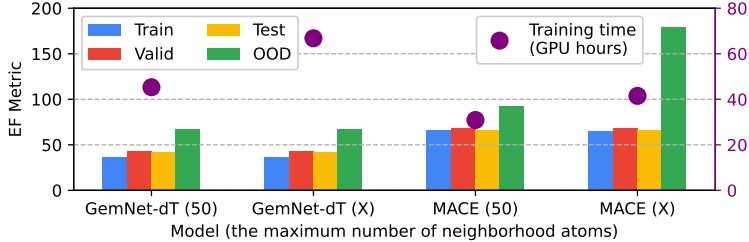

Figure C.1: EF metrics and training times of GemNet-dT and MACE trained with or without restricting the number of neighborhood atoms.

We trained models using a linear decaying schedule (LinearLR) that an learning rate is linearly decayed every step until the training is completed, *i.e.*, 200 epochs. The most existing MLFF models primarily employed one of the two types of learning rate schedules: ReduceLROnPlateau [19, 22, 23, 24] and StepLR [17, 18, 25] implemented in PyTorch. The former stably gives decent training results but makes it difficult to estimate when the training will be completed. The latter allows for predicting the training completion point by specifying the training cost. However, it requires careful consideration of factors such as the decaying points, the decay factor, and the number of decay steps. Therefore, to control the training cost, we opted for a linear decaying schedule with solely one required hyperparameter, an initial learning rate, and thus we can control the training budget for fair benchmark. During training, the learning rate linearly decreases from the initial learning rate to 0 at each iteration.

### C.3 Data Preprocessing

#### C.3.1 Graph Generation

For the fairness comparison among MLFF models, we trained MLFF models with a **radius cutoff of 6.0** and a maximum limit of **50 neighborhood atoms**. As illustrated in Figure C.1, despite the 1.5x and 1.3x increase in training costs for GemNet-dT and MACE, respectively, the test EF metrics of the two models are hardly improved. This implies that the atomic features, derived by message passing in GNNs, may be more strongly influenced by the closer neighborhood atoms. In MACE, utilizing more neighborhood atoms does not lead to improvements on the OOD set, but instead results in higher EF metric. Thus, analyzing the effect of graph generation conditions can be important to obtain models that have lower OOD errors and are more suitable for simulations.

#### C.3.2 Normalization for Energy and Forces

In this benchmark, when training MLFF models, we adopt a common normalization strategy for energy and forces. Here, we introduce normalization for energy and forces, and show our empirical results that present little difference regardless of a normalization strategy.

Normalizing energy and forces is commonly utilized in MLFF research, either through explicit normalization of energy and forces [29] or by incorporating scale and shift factors within the model to fulfill the normalization role [22, 23, 24]. In this Section, after defining the normalization of energy and forces, we analyze the effect of the normalization to training results.

First, to introduce the normalization technique for energy and forces, we have to be aware that the force of an atom is equal to the derivative of total energy with respect to the atom position in various MLFF models. Calculating forces via taking the derivatives naturally induces two properties: sum of all forces in a single snapshot is equal to $\mathbf{0}$, and multiplying a constant to an energy affects exactly same to the forces. Fortunately, the first property implies that the forces does not require centralization; *i.e.*, the forces are not shifted because their mean is $\mathbf{0}$. Thus, it suffices to scale the forces by dividing their standard deviation $\sigma_f$ [22, 23, 24, 29]. Moreover, by the second property, the scaling factor of the energy should be identical to the scaling factor of the forces, *i.e.*, $\sigma_f$.

The only remaining part in defining the normalization is how to centralize the energy. Several methods for the centralization have been proposed. As discussed about error metrics and losses, the energy types to be targeted for obtaining the shift factor are two: total energy and per-atom energy.

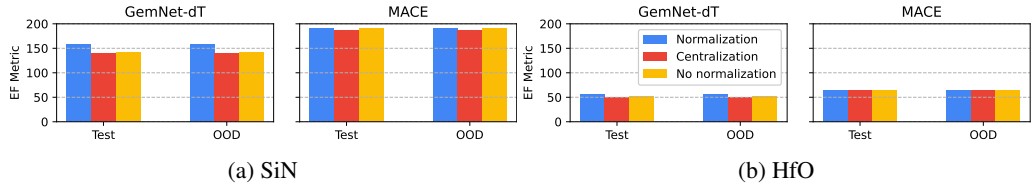

Figure C.2: EF metrics of GemNet-dT and MACE trained with different normalization strategies.

The total energy can be centralized as $E_i - \bar{E}$, where $E_i$ is the total energy of snapshot $i$ and $\bar{E}$ is the mean of total energies [29]. However, for MLFF models that calculate the total energy by predicting and summing up individual atomic energies, the centralization of total energy only works properly for the datasets, which consist of snapshots with the identical size.

As an alternative, some researches [22, 23, 24] centralize per-atom energy instead of the total energy, which is defined as

$$E_i - \frac{n_i}{N} \sum_i \frac{E_i}{n_i}, \tag{2}$$

where $n_i$ is the size of snapshot $i$ and thus $\frac{1}{N} \sum_i \frac{E_i}{n_i}$ is the mean of per-atom energies. When shifting the total energy, the mean of per-atom energies is compensated by multiplying the snapshot size. Since the SiN dataset consists of various snapshots with different number of atoms, Eq. 2 is employed for the centralization. Finally, the normalization of energy and forces for this benchmark is formulated as

$$\tilde{E}_i = \left( E_i - \frac{n_i}{N} \sum_i \frac{E_i}{n_i} \right) \Big/ \sigma_f , \qquad \tilde{\mathbf{F}}_i = \mathbf{F}_i / \sigma_f \tag{3}$$

MACE [24] can use per-species normalization, which is an extension of the per-atom normalization, by computing per-species energy for each chemical species by solving linear system set by the datasets. However, such method is not appropriate to the datasets which only contain snapshots with uniform stoichiometric ratio, such as the HfO dataset; the linear system becomes singular, and thus the per-species energy is not determined.

To explore the influence of the normalization strategies on the training results, we trained models using three different strategies: per-atom normalization, per-atom centralization, and no normalization. As presented in Figure C.2, the differences in errors obtained from models trained using these three methodologies are marginal, even in OOD errors. The energy prediction method employed by the MLFF models in this benchmark entails obtaining per-atom values within the model, which are then summed up to calculate the total energy of a snapshot. Thus, even not employing normalization, internal representations in MLFF models can inherently aligns with the per-atom values and be used in simulations for structures whose sizes are different from those in the training dataset. This is empirically demonstrated by the small error differences between test and OOD data.

### C.4 Supplementary Results

### C.4.1 Results of Benchmark for Energy and Force

Table C.3 presents numerical results, which are visualized in Figure 1. As in discussed in Section 4.1, we trained MLFF models with two loss functions. For SiN and HfO datasets, we illustrate the training results of these two loss functions in Figure C.3 (a) and (b), respectively; blue and red bars represent the training results of MSE- and MAE-based loss functions, respectively. Section 4.2 covers the analysis about these results.

In addition, maintaining the same training recipes except setting the number of epochs as 50, we trained the models using OC20 dataset [29]; the training results are visualized in Figure C.3 (c). Training recipes of OC20 benchmark are different from ours. For instance, the target energy values of snapshots are adjusted by subtracting so-called reference energy, which corresponds to the energy of an initial snapshot with the same trajectory as the snapshots, and then apply the normalization technique for the shifted energy values; in contrast, we use the target energy values and apply per-atom energy normalization. Furthermore, DimeNet++ and GemNet-dT used for OC20 benchmark

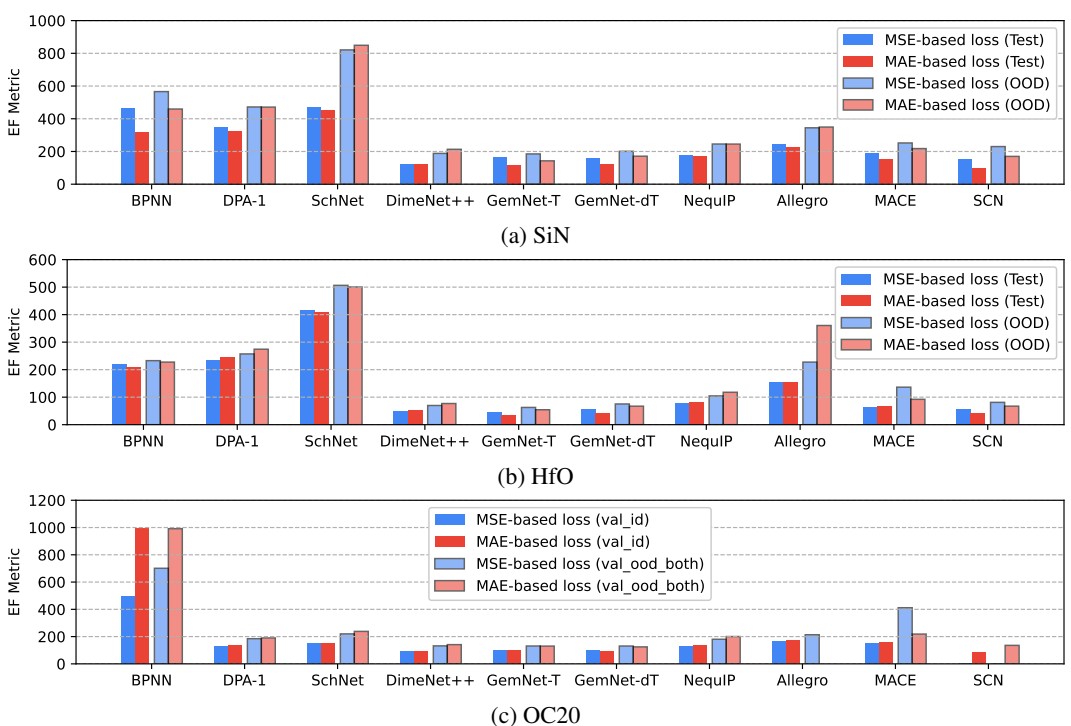

Figure C.3: EF metrics of 10 MLFF models trained with MSE- and MAE-based loss functions.

have larger architectures than those introduced in their original papers, while we utilize the original architectures; nevertheless, they show prominent results in energy and force prediction.

Overall, GNN-based models experience high-ranked performance. GNNs incorporating equivariance, such as Allegro and MACE, demonstrate inferior performance in energy and force prediction compared to SchNet, which is already considered the least favorable model in SiN and HfO datasets. For models except BPNN and Allegro, differences between the training results of two loss functions are less than those in SiN and HfO datasets, and EF metrics of using MAE-based loss are very slightly higher than using MSE-based loss. The results obtained from BPNN, where EF metrics of the model trained with MAE-based loss on both val_id and val_ood_both are higher, imply that descriptor-based models may experience a significant performance drop, when dealing with datasets containing a wide range of atom species such as OC20. Allegro trained with MAE-based loss may be reasonably trained when observing the results of val_id, but fails to correctly predict energy and forces on OOD data of OC20, referred to as val_ood_both. It may be beneficial to analyze which factors of Allegro bring about such failure, because Allegro is compatible with the parallel computing of MD simulation tools such as LAMMPS while the other models cannot, which is important to enable large-scale simulations using MLFF models.

### C.4.2 Results of Model Exploration Study

The numerical results of model exploration study, which are discussed in Section 4.3 and visualized in Figure 3, are listed in Table C.4, where the variation scales of each model and corresponding model sizes (*i.e.*, the number of parameters) are also included. The result of DPA-1 shows that DPA-1 can locate on the pareto-frontier in Figure 3. DPA-1, among the models evaluated in this study, is the only model that can be operated exclusively in DeePMD-kit [48], a TensorFlow-based framework. Therefore, the result of DPA-1 was excluded from Figure 1 because it may not be appropriate to directly compare that with the results of other models obtained from our PyTorch-based framework.

### C.5 Data Scaling Effect

We additionally present the data scaling effect, which may be helpful to make a training strategy for MLFF researchers. We follow the experimental settings of previous works [22, 24], where the

Table C.3: Numerical results (EF metrics) of MLFF models, visualized in Figures 1 and C.3.

| Model | Loss | SiN | | | HfO | | | OC20 | | |
|---|---|---|---|---|---|---|---|---|---|---|
| | | Train | Test | OOD | Train | Test | OOD | train | val_id | val_ood_both |
| BPNN | MSE-based | 393.3 | 466.2 | 566.0 | 219.2 | 220.5 | 232.7 | 477.2 | 495.0 | 701.1 |
| | MAE-based | 291.4 | 316.2 | 459.4 | 206.4 | 210.2 | 227.4 | 984.2 | 991.3 | 991.5 |
| DPA-1 | MSE-based | 335.0 | 347.1 | 472.0 | 232.7 | 234.0 | 257.1 | 127.8 | 128.9 | 184.6 |
| | MAE-based | 309.9 | 325.5 | 471.2 | 246.2 | 247.0 | 274.3 | 131.1 | 132.9 | 190.0 |
| SchNet | MSE-based | 192.0 | 471.2 | 820.7 | 171.5 | 417.4 | 506.4 | 145.3 | 147.2 | 219.8 |
| | MAE-based | 114.6 | 453.5 | 849.1 | 190.8 | 408.3 | 500.6 | 152.1 | 153.1 | 238.2 |
| DimeNet++ | MSE-based | 84.3 | 119.8 | 188.3 | 37.1 | 48.0 | 69.7 | 102.1 | 88.2 | 131.9 |
| | MAE-based | 64.8 | 124.1 | 213.3 | 42.8 | 51.6 | 76.9 | 108.8 | 93.0 | 141.1 |
| GemNet-T | MSE-based | 144.5 | 164.9 | 184.9 | 43.0 | 45.8 | 62.7 | 113.4 | 102.4 | 130.9 |
| | MAE-based | 98.2 | 118.1 | 142.7 | 27.3 | 33.0 | 54.3 | 112.1 | 102.0 | 130.1 |
| GemNet-dT | MSE-based | 138.5 | 158.5 | 201.0 | 53.4 | 56.8 | 75.2 | 110.3 | 98.3 | 130.7 |
| | MAE-based | 87.9 | 120.4 | 171.1 | 36.3 | 41.9 | 67.2 | 103.4 | 91.4 | 124.5 |
| NequIP | MSE-based | 163.0 | 174.4 | 245.5 | 75.1 | 79.6 | 104.5 | 133.9 | 126.3 | 180.4 |
| | MAE-based | 157.9 | 170.4 | 245.1 | 79.8 | 82.6 | 117.8 | 140.7 | 135.0 | 200.4 |
| Allegro | MSE-based | 236.2 | 245.1 | 344.6 | 152.5 | 156.2 | 227.4 | 168.9 | 163.7 | 213.7 |
| | MAE-based | 209.0 | 225.4 | 349.3 | 152.7 | 156.0 | 360.5 | 169.4 | 169.6 | 2.1e+14 |
| MACE | MSE-based | 186.7 | 190.3 | 252.0 | 62.8 | 64.6 | 136.3 | 153.2 | 146.8 | 411.8 |
| | MAE-based | 147.1 | 155.1 | 217.7 | 65.7 | 66.2 | 92.2 | 164.5 | 158.5 | 218.4 |
| SCN | MSE-based | 138.2 | 153.5 | 230.0 | 50.4 | 56.9 | 81.1 | - | - | - |
| | MAE-based | 50.2 | 97.9 | 170.0 | 30.1 | 41.5 | 67.3 | 105.5 | 87.7 | 135.6 |

ReduceLROnPlateu scheduler is used to train MLFF models. Even though the training steps cannot be fixed due to this scheduler unlike our setting, we anticipate that their training might be stopped at similar update steps. Thus, in this data scaling experiments, MLFF models are trained by the fixed number of steps for model update; the setting is referred to as the equal budget setting. We randomly sample 20%, 40%, 60%, and 80% snapshots from the HfO training set (*i.e.*, 5.6k, 11.2k, 16.8k, and 22.4k), and trained models by 1000, 500, 334, and 250 epochs, correspondingly, according to the equal budget setting. We select six models (BPNN, GemNet-T, GemNet-dT, NequIP, Allegro, and MACE) and obtain their energy and force errors using the HfO test and OOD sets. Except the data size and epochs, the training recipe used in the MAE-based training experiments of Figure 1 is maintained.

The results are reported in Table C.5 and visualized in Figure 4. For all models, as more training data are used, the training error increases while both the test error decreases, where the gap between these two errors also decrease. The OOD error decreases in most cases. Therefore, our data scaling results suggest that training with fewer epochs and more data can be helpful to improve the generalization performance of MLFF models. Meanwhile, the difference between EF metrics using 80% and 100% of training set is marginal, meaning that the performance improvement that can be gained by using more data seems to be saturated. Such observation implies that our training set sampled from the raw dataset is sufficient for our semiconductor MLFF benchmark.

We can opt for another data scaling experimental setup where the identical training epochs are used to train MLFF models; the setting is referred to as the equal epoch setting. In the equal epoch setting, as the training set size is reduced, the number of model update steps is also reduced. Considering that accuracy drop was clearly observed when using less data at the equal budget setting, we omit the experiments of the equal epoch setting because further accuracy drop is intuitively expected.

# D    Details of Properties Benchmark

## D.1    Radial and Angular Distribution Functions

The dynamic indicators, namely RDF and ADF, are classified as such since they are derived from high-temperature MD simulation trajectories. In order to investigate the stability of models in different atomic environments induced by high turbulence or active movement caused by high thermal energy,

Table C.4: Numerical results visualized in Figure 3 (HfO test set). Double, Base, Half, and Quarter indicate the range of the relative variation scale {2x, 1x, 0.5x, 0.25x}, correspondingly. [†]DPA-1 is specifically designed for use with DeePMD-kit, whereas the other models are implemented and evaluated within our framework.

| Model | Variation scale | Number of Params. | EF metric | Inference time (ms) |
|---|---|---|---|---|
| BPNN | Double | 1.92M | 356.9 | 31.2 |
| | Base | 0.50M | 356.1 | 31.5 |
| | Half | 0.13M | 358.9 | 32.1 |
| DPA-1[†] | Base | 6.14M | 234.0 | 23.9 |
| SchNet | Double | 35.61M | 743.6 | 24.3 |
| | Base | 9.09M | 687.8 | 23.4 |
| | Half | 2.37M | 627.0 | 23.1 |
| | Quarter | 0.64M | 692.3 | 24.8 |
| DimeNet++ | Base | 1.89M | 81.8 | 62.5 |
| | Half | 0.48M | 111.6 | 68.3 |
| | Quarter | 0.13M | 145.2 | 68.9 |
| GemNet-T | Base | 1.89M | 51.1 | 64.3 |
| | Half | 0.48M | 67.6 | 62.1 |
| | Quarter | 0.13M | 94.5 | 62.4 |
| GemNet-dT | Double | 9.14M | 55.0 | 34.5 |
| | Base | 2.31M | 64.7 | 29.5 |
| | Half | 0.59M | 84.7 | 24.9 |
| NequIP | Double | 1.45M | 113.9 | 62.1 |
| | Base | 0.36M | 133.4 | 62.0 |
| | Half | 0.09M | 160.0 | 61.3 |
| | Quarter | 0.02M | 197.5 | 57.0 |
| Allegro | Double | 5.61M | 243.1 | 48.7 |
| | Base | 1.40M | 263.1 | 30.0 |
| | Half | 0.35M | 289.9 | 27.4 |
| MACE | Double | 0.26M | 86.1 | 35.6 |
| | Base | 0.11M | 104.3 | 36.2 |
| | Half | 0.05M | 128.3 | 35.9 |
| SCN | Base | 1.58M | 62.2 | 327.2 |
| | Half | 0.47M | 90.9 | 316.2 |
| | Quarter | 0.13M | 183.3 | 332.2 |

we carefully selected high temperatures for SiN (1200 K) and HfO (1200 and 1800 K). RDF, also known as the pair correlation function, captures the changes in density relative to the distance from a chosen reference particle. On the other hand, ADF expands the scope of analysis beyond radial distances and centers on characterizing the angular distribution of particles surrounding a reference particle. The analysis of RDF and ADF plays a fundamental role in simulations as it enables the identification of structures, phases, and interactions. Moreover, it provides a deeper understanding of behaviors and reactions occurring within solids.

The radial distribution function, $g(r)$ is an average density of particles within distance $r$, which is represented by

$$g(r) = \frac{\mathrm{d}n_r}{4\pi\rho r^2 \mathrm{d}r},\tag{4}$$

where $\rho$ indicates an average density of the system.

For compound AB, there could be three types of RDF results: A-A, A-B, and B-B. Mean Absolute Error (MAE) was calculated against the DFT reference for each case. The resulting three MAE values were then averaged to obtain the RDF error for each specific structure. Similarly, ADF could yield six types of results: A-A-A, B-B-B, A-B-A, B-A-B, A-A-B, and B-B-A. For each ADF case, the

Table C.5: Numerical results visualized in Figure 4.

| Model | Training set size (ratio) | EF Metric ($\downarrow$) | | | Model | Training set size (ratio) | EF Metric ($\downarrow$) | | |
| --- | --- | --- | --- | --- | --- | --- | --- | --- | --- |
| | | Train | Test | OOD | | | Train | Test | OOD |
| BPNN | 5.6k (20%) | 190.1 | 231.4 | 260.6 | NequIP | 5.6k (20%) | 72.5 | 99.4 | 174.6 |
| | 11.2k (40%) | 200.9 | 216.4 | 239.1 | | 11.2k (40%) | 76.6 | 88.1 | 148.5 |
| | 16.8k (60%) | 204.1 | 212.9 | 232.4 | | 16.8k (60%) | 78.1 | 84.2 | 130.0 |
| | 22.4k (80%) | 205.8 | 210.8 | 230.8 | | 22.4k (80%) | 79.0 | 82.5 | 119.1 |
| | 28.0k (100%) | 206.4 | 210.2 | 227.4 | | 28.0k (100%) | 79.8 | 82.6 | 117.8 |
| GemNet-T | 5.6k (20%) | 19.8 | 48.9 | 75.2 | Allegro | 5.6k (20%) | 144.7 | 174.6 | 1923.8 |
| | 11.2k (40%) | 23.6 | 39.3 | 64.7 | | 11.2k (40%) | 148.9 | 163.8 | 658.3 |
| | 16.8k (60%) | 25.1 | 35.2 | 59.2 | | 16.8k (60%) | 150.8 | 157.5 | 365.8 |
| | 22.4k (80%) | 26.0 | 33.3 | 54.1 | | 22.4k (80%) | 152.0 | 156.4 | 415.3 |
| | 28.0k (100%) | 27.3 | 33.0 | 54.3 | | 28.0k (100%) | 152.7 | 156.0 | 360.5 |
| GemNet-dT | 5.6k (20%) | 25.1 | 62.4 | 91.8 | MACE | 5.6k (20%) | 61.7 | 71.0 | 221.6 |
| | 11.2k (40%) | 31.7 | 49.0 | 79.2 | | 11.2k (40%) | 64.3 | 68.3 | 180.4 |
| | 16.8k (60%) | 34.4 | 44.4 | 73.0 | | 16.8k (60%) | 65.1 | 67.6 | 126.0 |
| | 22.4k (80%) | 35.3 | 42.1 | 67.0 | | 22.4k (80%) | 66.4 | 67.3 | 114.6 |
| | 28.0k (100%) | 36.3 | 41.9 | 67.2 | | 28.0k (100%) | 66.1 | 66.3 | 92.2 |

MAE was calculated by comparing it to the respective DFT reference. The resulting six MAE values were then averaged to obtain the ADF error for each specific structure.

To compare two distribution functions, we employ L1 distance between distribution functions. Thus an error metric of a RDF $g(r)$ generated by a MLFF is defined by

$$E_{\text{RDF}} = d(g, g_{\text{DFT}}) = \frac{1}{R_c} \int_0^{R_c} |g(r) - g_{\text{DFT}}(r)| \mathrm{d}r \tag{5}$$

where, $R_c$ is equal to 6Å.

Similarly, an error of an ADF $h(\theta)$ is defined by

$$E_{\text{ADF}} = d(h, h_{\text{DFT}}) = \frac{1}{\pi} \int_0^{\pi} |h(\theta) - h_{\text{DFT}}(\theta)| \mathrm{d}\theta. \tag{6}$$

In both datasets, high-temperature simulations with various initial structures were conducted for a duration of 6 ps. Ground truth trajectories were computed using DFT. RDF and ADF were computed by averaging the last 3 ps of the simulation; a total of 50 snapshots were averaged, with an interval of 6 fs between each snapshot. The simulated structures employed in this study were supercells, encompassing a varying number of atoms. Specifically, the supercells comprised a range of 2,592 to 3,456 atoms for HfO and 1,296 to 2,835 atoms for SiN (refer to Table D.1 for additional information). This range of atom counts facilitates future scalability studies of MLFF. Additionally, users have the flexibility to increase supercell sizes as desired.

For the SiN, four different structures were simulated at 1200 K. Two structures with Si:N ratios of 3:4 and 1:1, which were within the range of ratios covered by the training dataset, were used to compute RDF$^{\text{ID}}$ and ADF$^{\text{ID}}$. The RDF$^{\text{ID}}$ value was calculated by averaging the RDF errors of the two structures, and likewise, the ADF$^{\text{ID}}$ value was obtained by averaging the ADF errors of the two structures. Additionally, two structures with Si:N ratios of 1:2 and 3:2, which were outside the range of ratios present in the training dataset, were employed to calculate RDF$^{\text{OOD}}$ and ADF$^{\text{OOD}}$.

A total of eight structures were simulated for the HfO. Among these, five structures were simulated at 1200 and 1800 K (resulting in a total of 10 simulations) using the same stoichiometry (Hf:O = 1:2) as in the training set. These simulation trajectories were then used to calculate RDF$^{\text{ID}}$ and ADF$^{\text{ID}}$. On the other hand, the remaining three structures were considered out-of-distribution structures with varying stoichiometries of Hf:O, specifically 1:1, 2:3, and 4:7. These three structures were simulated at 1200 K and then employed to calculate RDF$^{\text{OOD}}$ and ADF$^{\text{OOD}}$.

### D.1.1 Understanding RDF & ADF: Insight, Motive, and Explanation

In simulation studies, understanding atomic position patterns can be complex. To solve this, researchers turn to RDF and ADF, two pivotal post-processing techniques, to differentiate between

Table D.1: Details of evaluated structures for RDF and ADF. Structures with index (idx.) indicate the same crystal family but have different lattice constants. The index numbering has been restarted for each dataset (e.g., Cubic (1) of SiN and Cubic (1) of HfO have different lattice constants).

| Data | Distribution | Structure (idx.) | Hf:O (or Si:N) | # atoms (3×3×3) | Temp. |
|---|---|---|---|---|---|
| SiN | ID | Amorphous | 1:1 | 1296 | 1200 K |
| | | Triclinic | 3:4 | 2835 | 1200 K |
| | OOD | Cubic (1) | 1:2 | 2592 | 1200 K |
| | | Cubic (2) | 3:2 | 2720 | 1200 K |
| HfO | ID | Monoclinic | 1:2 | 2592 | 1200 & 1800 K |
| | | Tetragonal | 1:2 | 2592 | 1200 & 1800 K |
| | | Cubic (1) | 1:2 | 2592 | 1200 & 1800 K |
| | | Orthorhombic (1) | 1:2 | 2592 | 1200 & 1800 K |
| | | Orthorhombic (2) | 1:2 | 2592 | 1200 & 1800 K |
| | OOD | Hexagonal (1) | 1:1 | 3456 | 1200 K |
| | | Hexagonal (2) | 2:3 | 3240 | 1200 K |
| | | Cubic (2) | 4:7 | 2376 | 1200 K |

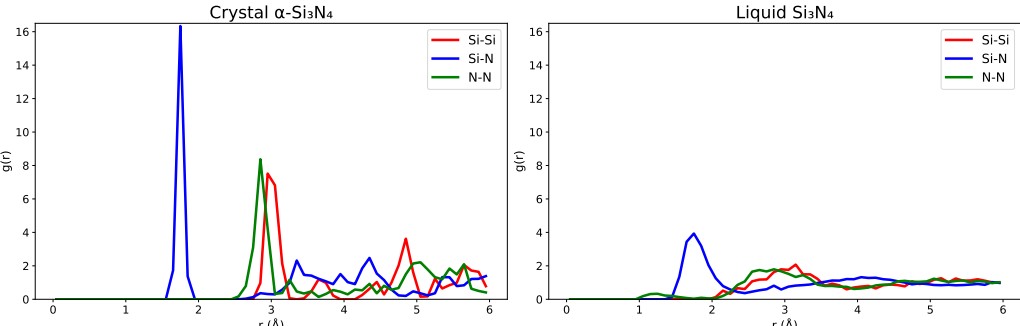

Figure D.1: RDFs of $\alpha$-$Si_3N_4$ (left) and liquid-$Si_3N_4$ (right).

materials or states. RDF quantifies atomic or molecular density fluctuations based on distance, providing insights into localized structural properties. In contrast, ADF elucidates angular tendencies between particle triplets, offering a perspective on molecular shapes and bonding angles. Collectively, they act as benchmarks, verifying the real-world alignment of simulations.

At the heart of MLFF models lies the ability to accurately predict energies and forces governing atoms and molecules. These predictions are crucial as they directly influence the movement, behavior, and interactions of atoms, which, in turn, dictate the patterns seen in RDF and ADF. When the RDF and ADF predicted by an MLFF closely align with those from DFT calculations, it's a strong indication that the model is capturing the essential physics and is making accurate energy and force predictions.

RDF and ADF can be used to capture phenomena such as phase transitions, system characteristics, and intermolecular dynamics. A prime example is the simulation of the silicon nitride phase shift from solid to liquid, where stark RDF modifications are observable in Figure D.1. What begins as distinct peaks in its crystalline form smoothes out during the melting process, signifying inherent structural shifts. Figure D.1 contrasts the RDF patterns of crystalline silicon during its initial state and post-melting phase, underlining the transformative effects of atomic rearrangements. Such configurations inherently relate to the balance of interatomic forces and the resultant energy landscape. When atoms approach each other under specific conditions, they might form bonds due to attraction or repel, settling into stable distances. This equilibrium gives rise to distinctive RDF and ADF profiles that mirror these atomic positions.

While RDF and ADF primarily help visualize atomic configurations, they also play a crucial role in assessing the fidelity of atomic paths predicted by MLFF. For example, in Figure D.3-(c), there is a noticeable anomaly in the RDF for Hf-Hf interactions on the OOD Hexagonal (2) structure when obtained using the MAE-based SchNet model. This anomaly is characterized by a pronounced peak at unusually short 'r' distances. This unlikely closeness between Hf-Hf atoms, especially when

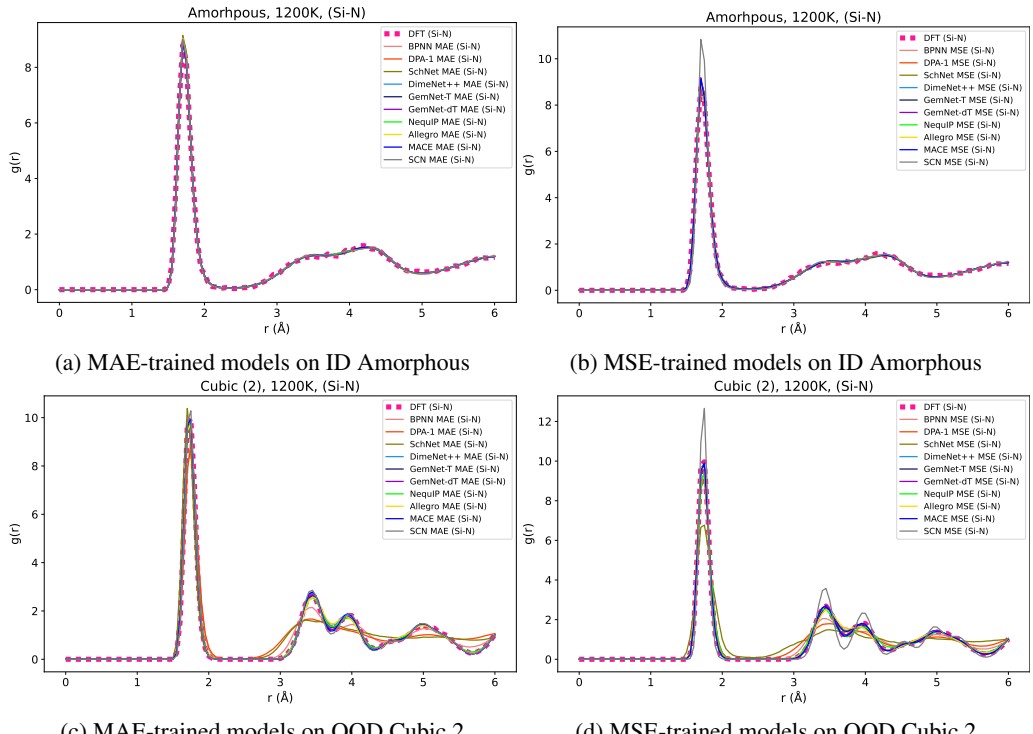

(a) MAE-trained models on ID Amorphous     (b) MSE-trained models on ID Amorphous

(c) MAE-trained models on OOD Cubic 2     (d) MSE-trained models on OOD Cubic 2

Figure D.2: RDF comparisons: DFT ground truth vs. model predictions for SiN in ID Amorphous and OOD Cubic 2 structures.

compared to ground truth data, not only challenges physical rationale but also signals potential weaknesses in the predictive capability of the MLFF model.

In our framework, we introduce straightforward evaluations of RDF and ADF for MLFF models by contrasting them against reference profiles from gold-standard DFT simulations (Table D.1). A close match between the MLFF-derived profiles and these references signifies model accuracy. Though a smaller error naturally indicates superior model performance, for visualization in the form of radar plots as shown in Figures D.13 and D.14, we inversely transform the RDF and ADF errors such that a value of 1 denotes optimal performance. The specifics of this transformation are detailed in Section D.4.

## D.2 Bulk Modulus and Equilibrium Volume

The bulk modulus and equilibrium volume derived from the Birch-Murnaghan equation of state (EoS) are critical in investigating thin film materials, including silicon nitride andd hafnium oxide. The bulk modulus provides insights into the mechanical stability of these films, indicating their resistance to volume shifts under varying pressure conditions and their resilience against external stresses. Additionally, it characterizes the elastic properties of the films, impacting aspects like flexibility and hardness. Meanwhile, the equilibrium volume aids in understanding the relationship between the film's thickness and residual stress, enabling us to evaluate the film's state (strained or relaxed), and refine the growth procedure. Furthermore, both these parameters influence the interfacial properties such as adhesion strength and interface energy, integral for the performance and stability of thin film systems. Comprehension of these parameters is essential for engineering thin films with the required properties, for their potential application in electronics, optoelectronics, and microelectromechanical systems.

The bulk modulus is a fundamental quantity that measures a material's resistance to compression and expansion. It quantifies the material's ability to withstand changes in volume when subjected to external pressure. Comparing the bulk moduli of different materials allows for the assessment of their relative compressibility and overall mechanical strength. Additionally, the equilibrium volume, obtained through the fitting of the Birch-Murnaghan EoS, represents the volume at which the material

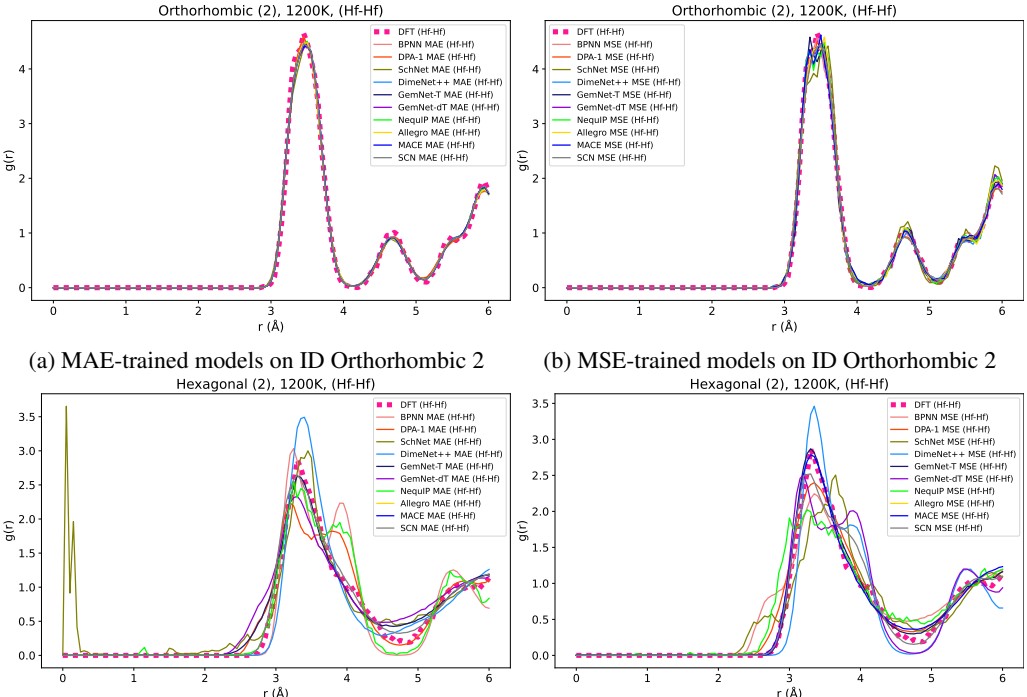

(a) MAE-trained models on ID Orthorhombic 2      (b) MSE-trained models on ID Orthorhombic 2

(c) MAE-trained models on OOD Hexagonal 2      (d) MSE-trained models on OOD Hexagonal 2

Figure D.3: RDF comparisons: DFT ground truth vs. model predictions for HfO in ID Orthorhombic 2 and OOD Hexagonal 2 structures.

possesses the minimum energy. Determining the equilibrium volume enables the identification of the most energetically stable configuration of the material, providing insights into its structural stability and phase transitions.

The bulk modulus and equilibrium volume can be obtained by fitting the Birch-Murnaghan EoS. The EoS establishes a relationship between the energy and volume of a solid material, and it can be expressed as follows:

$$E\left(V\right) = E_0 + \frac{9V_0 B_0}{16}\left\{\left[\left(\frac{V_0}{V}\right)^{2/3} - 1\right]^3 B_0' + \left[\left(\frac{V_0}{V}\right)^{2/3} - 1\right]^2\left[6 - 4\left(\frac{V_0}{V}\right)^{2/3}\right]\right\}, \quad (7)$$

where $V_0$ is the equilibrium volume, $B_0$ is the bulk modulus, and $B_0'$ is the derivative of the modulus with respect to pressure.

The internal energy versus volume data obtained from DFT calculations served as reference to assess the accuracy of MLFF models. In the case of HfO, five crystalline structures present in the training set were utilized to calculate $B_0^{ID}$ and $V_0^{ID}$. Additionally, one crystal with different stoichiometry, not included in the training set, was employed to determine $B_0^{OOD}$ and $V_0^{OOD}$. Similarly, in the case of SiN, five different crystalline structures with varying numbers of atoms were used in the EoS evaluation. Among these structures, three were included in the training set and employed to calculate $B_0^{ID}$ and $V_0^{ID}$. The remaining two structures, which was not commonly found in the training set, was utilized to determine $B_0^{OOD}$ and $V_0^{OOD}$. For more detailed information regarding the structures evaluated please refer to Table D.2. The values of $B_0^{ID}$, $B_0^{OOD}$, $V_0^{ID}$, and $V_0^{OOD}$ were determined by calculating the arithmetic mean of the absolute percentage errors for the evaluated structures.

### D.2.1    Understanding the Equation of State: Insight, Motive, and Explanation

Our paper introduces the equation of state as an evaluation method, with metrics derived from the EoS, specifically the bulk modulus ($B_0$) and equilibrium volume ($V_0$), serving as the primary assessment metrics. These metrics offer a way to assess how closely the predictions of the MLFF models align with the results from more established methods, like DFT.

Table D.2: Details of evaluated structures for EoS.

| | Distribution | Space Group | Hf-O (or Si-N) ratio | Number of atoms |
|---|---|---|---|---|
| SiN | ID | P31c | 3:4 | 28 |
| | | P1 | 3:4 | 14 |
| | | P1 | 3:4 | 28 |
| | OOD | $Fd\bar{3}m$ | 3:4 | 14 |
| | | $Fd\bar{3}m$ | 3:4 | 56 |
| HfO | ID | $P2_1/c$ | 1:2 | 96 |
| | | $P4_2/nmc$ | 1:2 | 96 |
| | | $Fm\bar{3}m$ | 1:2 | 96 |
| | | Pbca | 1:2 | 96 |
| | | Pnma | 1:2 | 96 |
| | OOD | $Fd\bar{3}m$ | 4:7 | 88 |

The bulk modulus provides insights into a material's resistance to compression. In simple terms, it helps us understand how much a material can be compressed. A higher value indicates that the material is less likely to change its volume under pressure. The equilibrium volume, on the other hand, tells us about the optimal volume where a material is most stable and uses energy most efficiently.

The core requirement of the MLFF model is its capability to simulate atomic interactions. When these interactions are observed at a larger scale, they manifest as properties such as the material's resistance to compression and its most stable form. These properties are effectively represented by metrics like the bulk modulus and equilibrium volume. This suggests that a model which can accurately predict atomic interactions is also likely to be proficient at predicting these larger-scale properties.

An important validation of the MLFF model's effectiveness is its alignment with DFT results. If our MLFF model's predictions are consistent with DFT-calculated metrics, it indicates that our model has a good grasp on the intricate interactions between atoms. For ML experts, the EoS metrics serve a dual purpose. Not only do they offer a straightforward and clear measure of the model's performance, but they also bridge the gap between intricate atomic simulations and tangible material properties, making the underlying science more accessible and understandable.

In the real experiment, when the mismatch of lattice constant ($V_0{}^{1/3}$) exceeds over 5% between materials, it becomes very difficult to synthesize the materials. Thus, a reasonable MLFF model should predict volume accurately within this range. Since the GGA-level of DFT calculation already contains volume error of 2-3%[58], the error on the lattice constant should be less than 3%, which suggest that error on $V_0$ ($|V_{0,DFT} - V_{0,MLFF}|/V_{0,DFT}$) be less than 0.1.

Our framework offers an easy-to-use evaluation and reference data for a diverse range of solid structures (Table D.2). It not only automates the calculation of the bulk modulus and the equilibrium volume but also provides comparative DFT results as a gold standard. As depicted in these Figures D.4 and D.5, less effective MLFF models may struggle in accurately predicting atomic interactions when there's a change in volume due to varying pressures. This misalignment in prediction can lead to deviations in the energy shift graph from the ideal DFT profile. Such discrepancies indicate the model's challenges in capturing energy variations associated with changes in the internal atomic configurations, highlighting potential inaccuracies in forecasting the material's mechanical properties. This approach gives us a broader perspective, enabling an assessment of MLFF's capabilities beyond merely energy and force predictions, and demonstrates its application in real-world simulation scenarios.

### D.3 Potential Energy Curves

Potential energy curves for two-body interatomic interactions were computed by manipulating the distance between a pair of atoms. Additionally, specific MLFF models, which incorporate many-body terms, were assessed with an array of molecular structures that encompassed more than just two atoms. These many-body potential energy curves were derived by adjusting particular distances or angles within these molecular structures. In these evaluations, the ground truth obtained from DFT

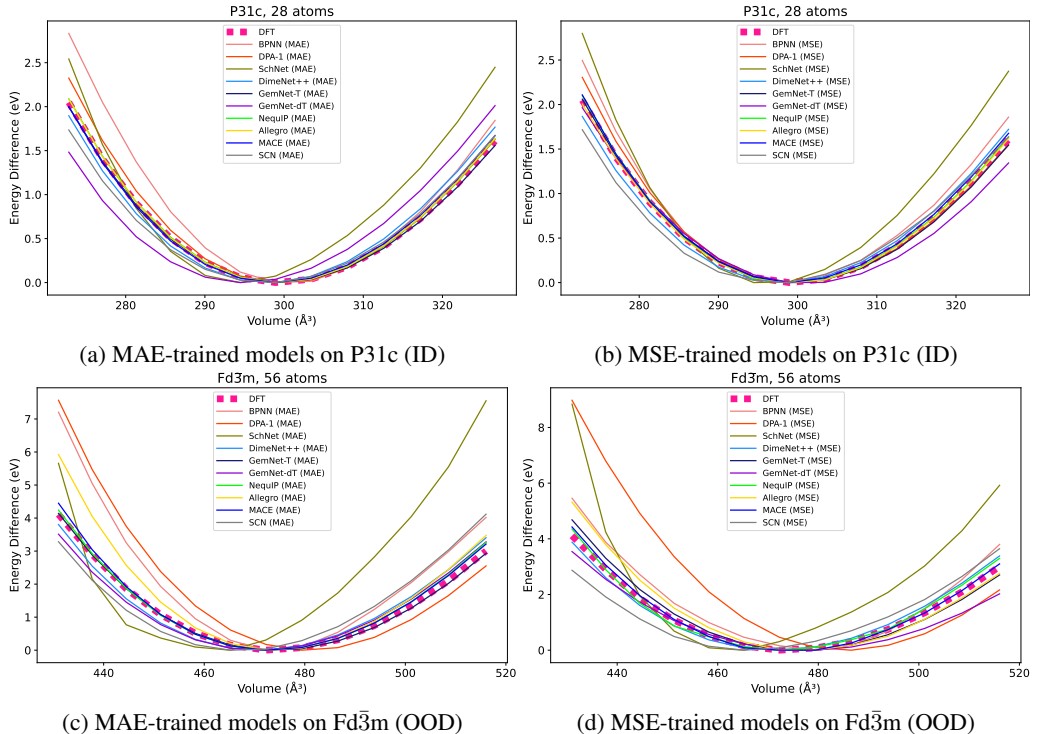

(a) MAE-trained models on P31c (ID)   (b) MSE-trained models on P31c (ID)

(c) MAE-trained models on Fd$\bar{3}$m (OOD)   (d) MSE-trained models on Fd$\bar{3}$m (OOD)

Figure D.4: EoS comparisons: ground truth (DFT) vs. model predictions on SiN P31c (ID) and Fd$\bar{3}$m (OOD) structures. Each curve is shifted to reflect the energy difference relative to its minimum value.

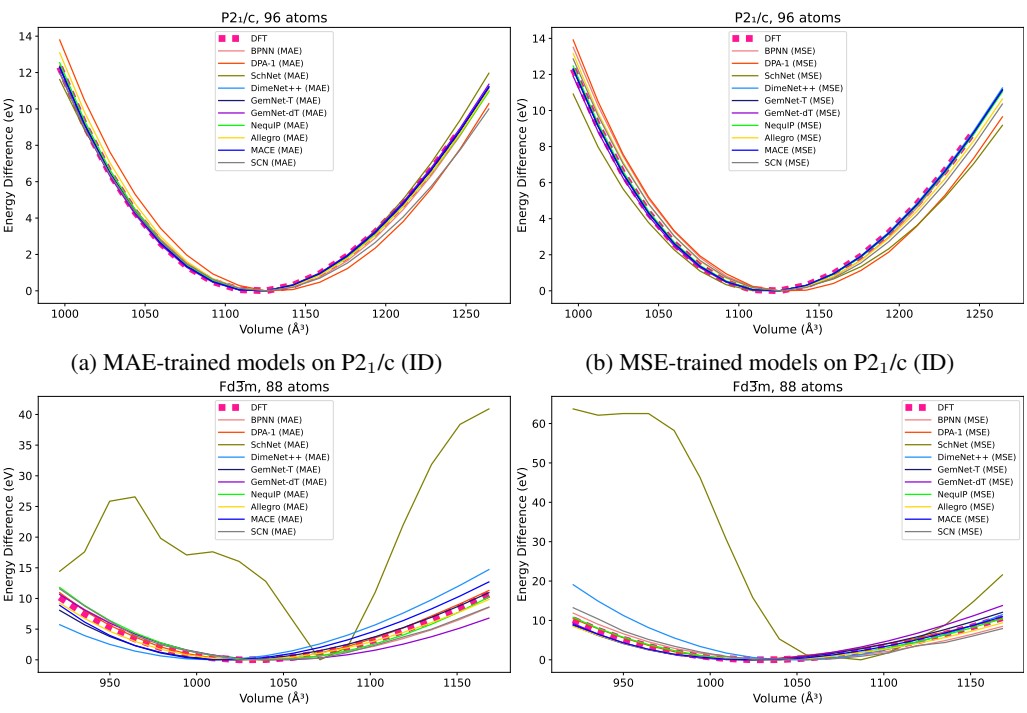

(a) MAE-trained models on P2$_1$/c (ID)   (b) MSE-trained models on P2$_1$/c (ID)

(c) MAE-trained models on Fd$\bar{3}$m (OOD)   (d) MSE-trained models on Fd$\bar{3}$m (OOD)

Figure D.5: EoS comparisons: ground truth (DFT) vs. model predictions on HfO P2$_1$/c (ID) and Fd$\bar{3}$m (OOD) structures. Each curve is shifted to reflect the energy difference relative to its minimum value.

Table D.3: Details of PEC evaluations.

|  |  | Formula | Variable* | Range |
|---|---|---|---|---|
| SiN | two-body | Si-Si | distance (r) | $1.000 \sim 6.000$ Å |
|  |  | Si-N | distance (r) | $1.000 \sim 6.000$ Å |
|  |  | N-N | distance (r) | $0.500 \sim 6.000$ Å |
|  | many-body | $Si_3N$ | distance (r) | $2.000 \sim 6.000$ Å |
|  |  | $SiN_4$ | distance (r) | $2.000 \sim 6.000$ Å |
| HfO | two-body | Hf-Hf | distance (r) | $1.000 \sim 6.000$ Å |
|  |  | Hf-O | distance (r) | $1.000 \sim 6.000$ Å |
|  |  | O-O | distance (r) | $0.500 \sim 6.000$ Å |
|  | many-body | $HfO_2$ | distance (r) | $0.986 \sim 6.786$ Å |
|  |  | $HfO_3$ | angle ($\theta$) | $0 \sim 140°$ |
|  |  | $HfO_4$ | angle ($\theta$) | $0 \sim 180°$ |

*Please refer Figure D.6

calculations was used as a reference. For detailed information on these evaluations, please refer to Table D.3 and Figure D.6. In the case of SiN many-body structures, we employed equilateral triangles ($Si_3N$) and tetrahedron ($SiN_4$) as they represent the atomic environment of each element with their first nearest neighbors in crystals. Additionally, the many-body structures of HfO were prepared by referencing a study on monohafnium oxide clusters, which is relevant for understanding defect sites in HfO thin films [59]. PEC evaluation has the potential to be a valuable tool in the solid-state domain, particularly in scenarios where certain chemical reactions or electrochemical properties are expected to take place in rare atomic environments.

### D.3.1 Understanding PEC: Insight, Motive, and Explanation

MLFFs have emerged as a significant tool in computational materials science. Through data-driven processes, they can precisely predict energies and forces within molecular configurations. Their performance is considered reliable when they align well with PECs from high-fidelity DFT calculations.

PECs have historically been essential for empirical interatomic potential model fitting. They have served as the cornerstone of empirical methodologies, used to understand energy changes based on varying atomic configurations. Given this foundational role of PECs, their alignment with modern MLFFs becomes crucial.

The alignment of an MLFF with high-fidelity DFT-derived PECs verifies its ability to capture complex interactions and its competence in modeling forces. Notably, forces are derived from the gradients of the molecular energy landscape. The ability to predict energy landscapes, as validated by its congruence with DFT-derived PECs, ensures that the resulting molecular simulations are trustworthy. In summation, the alignment between MLFFs and DFT-derived PECs, especially across diverse molecular configurations, demonstrates the adaptability and effectiveness of the MLFF model.

A distinguishing feature of our framework is its user-friendly evaluation capability across a broad spectrum of molecular structures (Figure D.6) for both systems, especially concerning PECs. While our main focus has been on condensed-phase systems, handling sparse molecular systems poses inherent challenges. These sparse systems act as outliers for models predominantly trained on condensed-phase datasets. Successfully obtaining accurate PECs for such outliers underlines the robustness and versatility of the MLFF approach.

### D.3.2 PEC Evaluation

Figures D.9 and D.10 visualize the PEC evaluation results of models for SiN data, while Figures D.11 and D.12 depict the results for HfO. PEC evaluation poses a significant challenge as it involves assessing sparse atomic environments that are rarely encountered by models trained on condensed-phase datasets. While it is not essential for models used in condensed-phase material research to

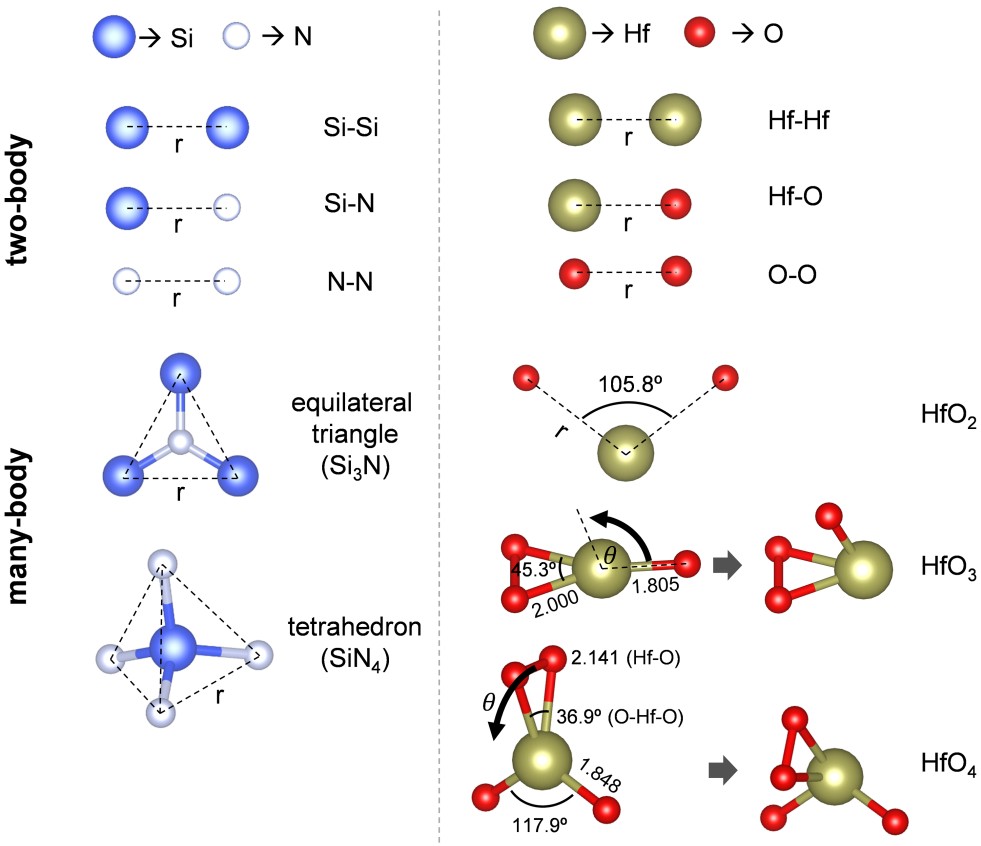

Figure D.6: Molecules used for PEC evaluations.

excel in PEC evaluation, the evaluation results provided valuable insights into the disparities between the two datasets. The SiN data included a wide range of atomic compositions and environments, including Si-only and N-only structures. In contrast, the HfO data exhibited a less diverse range. Consequently, the PEC evaluation results for SiN showed a closer resemblance to the DFT ground truth curve compared to the results for HfO.

To explain in a slightly more physics-oriented manner, given that most models predominantly train on low-energy data, predicting the high-energy PE surface becomes a challenging extrapolation task for these models. However, predicting this high-energy PE regime, usually occurring when the atoms get close, is important for stable MD simulation.

Two distinct methods can address this discrepancy in PE surface prediction. The first, termed as a data-centric approach, emphasizes the incorporation of more data from the high-energy regime, necessitating careful DFT calculations. The second approach, so called $\delta$-learning, tries to combine classical force-field model and MLFF models [60, 61, 62]. With this method, an initial approximation of the PE surface is derived using the classical force-field model, followed by the refinement of the residual portion using an MLFF model. Both strategies present potential solutions when confronting inconsistencies in the PE surface predictions.

### D.4 Calculating Indicator Score: Formula and Details

For all simulation indicators and numerical metrics, lower values indicate better results than higher values. To draw radar plots for model comparison, an inverse linear transformation were applied to map them between 0 and 1, where a score of 1 incidates the best performance. Minimum observed error was considered as the perfect score of 1, serving as the ideal benchmark for each metric. In order to maintain reasonable values, we meticulously selected maximum thresholds for each metric. For detailed information regarding the transformation rule and maximum thresholds, please refer to

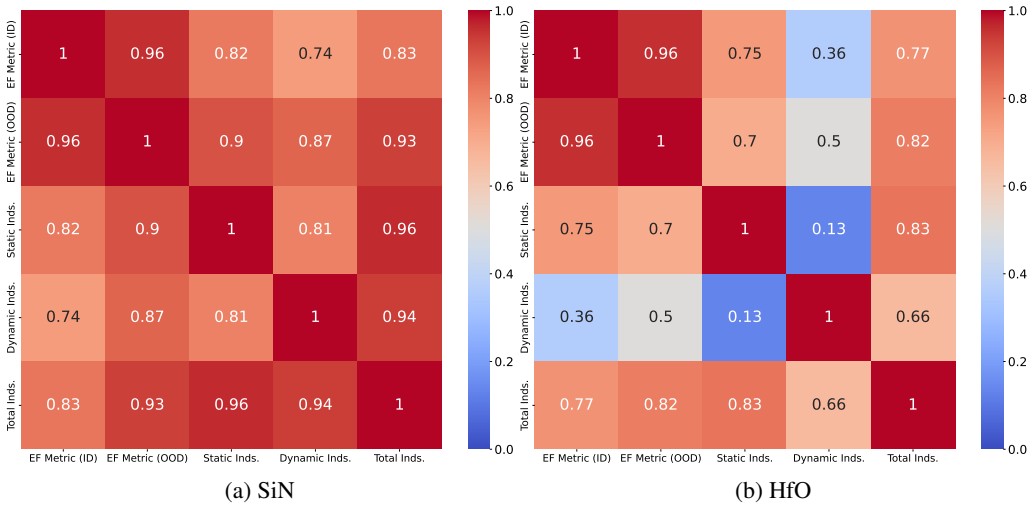

|          | (a) SiN | (b) HfO |
| --- | --- | --- |

Figure D.7: Pearson correlation matrix illustrating the relationships between "EF Metric (ID)", "EF Metric (OOD)", "Static Inds.", "Dynamic Inds.", and "Total Inds." across two datasets: (a) SiN and (b) HfO. The "Static Inds." and "Dynamic Inds." for each model were computed by summing the indicator scores from both ID and OOD cases. The "Total Inds." represents the combined sum of the "Static Inds." and "Dynamic Inds." scores.

Table D.4: Raw scores of models for indicators using SiN dataset and trained with MAE-based loss. Lower values indicate better performance.

|          | EF mtr.$^{\text{ID}}$ | EF mtr.$^{\text{OOD}}$ | RDF$^{\text{ID}}$ | RDF$^{\text{OOD}}$ | ADF$^{\text{ID}}$ | ADF$^{\text{OOD}}$ | $V_0^{\text{ID}}$ | $V_0^{\text{OOD}}$ | $B_0^{\text{ID}}$ | $B_0^{\text{OOD}}$ |
| --- | --- | --- | --- | --- | --- | --- | --- | --- | --- | --- |
| BPNN      | 3.2e+2 | 4.6e+2 | 4.7e-2 | 1.5e-1 | 3.5e-2 | 1.1e-1 | 5.4e-1 | 1.9e-1 | 2.1e+1 | 5.1e+1 |
| DPA-1     | 3.3e+2 | 4.7e+2 | 5.0e-2 | 1.9e-1 | 3.4e-2 | 2.2e-1 | 1.6e-1 | 1.5e+0 | 6.0e+0 | 2.5e+1 |
| SchNet    | 4.5e+2 | 8.5e+2 | 6.0e-2 | 1.1e+0 | 3.9e-2 | 4.3e-1 | 8.1e-1 | 1.9e+0 | 4.8e+1 | 9.3e+1 |
| DimeNet++ | 1.2e+2 | 2.1e+2 | 4.2e-2 | 1.1e-1 | 3.2e-2 | 9.2e-2 | 2.8e-1 | 5.7e-1 | 1.4e+0 | 2.7e+0 |
| GemNet-T  | 1.2e+2 | 1.4e+2 | 4.0e-2 | 3.3e-2 | 3.0e-2 | 2.1e-2 | 1.5e-1 | 1.1e-1 | 3.3e+0 | 1.2e+0 |
| GemNet-dT | 1.2e+2 | 1.7e+2 | 4.1e-2 | 6.9e-2 | 3.1e-2 | 5.9e-2 | 1.2e+0 | 5.8e-1 | 1.7e+0 | 3.2e+0 |
| NequIP    | 1.7e+2 | 2.5e+2 | 4.1e-2 | 7.0e-2 | 3.1e-2 | 4.1e-2 | 8.3e-2 | 1.4e-1 | 1.4e+0 | 5.2e+0 |
| Allegro   | 2.3e+2 | 3.5e+2 | 4.4e-2 | 4.2e-1 | 3.3e-2 | 2.3e-1 | 2.5e-2 | 8.2e-2 | 2.7e+0 | 2.8e+1 |
| MACE      | 1.6e+2 | 2.2e+2 | 4.0e-2 | 4.4e-2 | 3.0e-2 | 3.5e-2 | 7.1e-2 | 1.2e-1 | 4.5e-1 | 6.9e+0 |
| SCN       | 9.8e+1 | 1.7e+2 | 4.1e-2 | 5.7e-2 | 3.0e-2 | 4.4e-2 | 4.5e-1 | 1.3e+0 | 5.7e+0 | 7.9e+0 |

Table D.8. The raw scores of the models prior to this transformation are summarized in Tables D.4 to D.7. The details of calculating the raw score for each indicator are explained in Sections D.1 to D.3, along with evaluated structures and a description of simulation environments. The performance comparison among models based on indicators after the reverse transformation to scores ranging from 0 to 1 is visualized using radar plots in Figures D.13 and D.14.

## D.5 Empirical Model Analysis

**BPNN** requires hand-crafted features but has simplest overall structure, which makes the model relatively fast and less accurate than recent SOTA models.

**DPA-1** has relatively low accuracy on the EF metric, but shows better performance on simulation indicators.

**SchNet** lies on the pareto-frontier as the fastest model, but the speed gain may not be enough to compensate overall accuracy drop compared to recent fast models such as Allegro or GemNet-dT.

**DimeNet++** and **GemNet-T** are models with similar base structure. They show a similar overall tendency: generally high accuracy but less accurate on the $V_0$ indicator, and having relatively slow inference speed. Overall, GemNet-T, which lies on the most accurate side of pareto-frontier, seems slightly better predictive performance than DimeNet++.

Table D.5: Raw scores of models for indicators using SiN dataset and trained with MSE-based loss. Lower values indicate better performance.

| | EF mtr.$^{ID}$ | EF mtr.$^{OOD}$ | RDF$^{ID}$ | RDF$^{OOD}$ | ADF$^{ID}$ | ADF$^{OOD}$ | $V_0^{ID}$ | $V_0^{OOD}$ | $B_0^{ID}$ | $B_0^{OOD}$ |
|---|---|---|---|---|---|---|---|---|---|---|
| BPNN | 4.7e+2 | 5.7e+2 | 5.0e-2 | 1.7e-1 | 3.3e-2 | 1.4e-1 | 4.7e-1 | 7.5e-1 | 1.2e+1 | 3.2e+1 |
| DPA-1 | 3.5e+2 | 4.7e+2 | 5.0e-2 | 1.2e-1 | 3.5e-2 | 1.5e-1 | 3.6e-1 | 2.5e+0 | 2.0e+0 | 3.6e+1 |
| SchNet | 4.7e+2 | 8.2e+2 | 5.7e-2 | 1.1e+0 | 3.5e-2 | 4.9e-1 | 6.8e-1 | 1.8e+0 | 5.1e+1 | 1.5e+2 |
| DimeNet++ | 1.2e+2 | 1.9e+2 | 4.5e-2 | 2.1e-1 | 3.1e-2 | 7.0e-2 | 1.9e-1 | 4.8e-1 | 5.0e-1 | 3.0e+0 |
| GemNet-T | 1.6e+2 | 1.8e+2 | 4.5e-2 | 3.6e-2 | 3.0e-2 | 2.7e-2 | 3.3e-1 | 5.3e-1 | 1.9e+0 | 1.2e+0 |
| GemNet-dT | 1.6e+2 | 2.0e+2 | 5.0e-2 | 1.1e-1 | 3.1e-2 | 7.2e-2 | 7.4e-1 | 8.0e-1 | 1.1e+1 | 2.3e+1 |
| NequIP | 1.7e+2 | 2.5e+2 | 4.5e-2 | 8.4e-2 | 3.1e-2 | 5.1e-2 | 1.6e-1 | 2.7e-1 | 2.4e+0 | 6.5e+0 |
| Allegro | 2.5e+2 | 3.4e+2 | 4.9e-2 | 1.4e-1 | 3.2e-2 | 1.1e-1 | 2.3e-1 | 7.2e-1 | 3.9e-1 | 9.8e+0 |
| MACE | 2.0e+2 | 2.6e+2 | 4.6e-2 | 3.7e-2 | 3.2e-2 | 2.4e-2 | 1.9e-1 | 6.3e-2 | 3.2e+0 | 6.0e+0 |
| SCN | 1.5e+2 | 2.3e+2 | 6.7e-2 | 1.2e-1 | 3.5e-2 | 1.3e-1 | 7.7e-1 | 1.3e+0 | 7.4e+0 | 5.8e+0 |

Table D.6: Raw scores of models for indicators using HfO dataset and trained with MAE-based loss. Lower values indicate better performance. N/A indicates an interrupted simulation for the dynamic indicator due to abnormal energy changes.

| | EF mtr.$^{ID}$ | EF mtr.$^{OOD}$ | RDF$^{ID}$ | RDF$^{OOD}$ | ADF$^{ID}$ | ADF$^{OOD}$ | $V_0^{ID}$ | $V_0^{OOD}$ | $B_0^{ID}$ | $B_0^{OOD}$ |
|---|---|---|---|---|---|---|---|---|---|---|
| BPNN | 2.1e+2 | 2.3e+2 | 4.9e-2 | 1.3e-1 | 4.3e-2 | 1.1e-1 | 3.4e-1 | 1.1e+0 | 7.5e+0 | 1.0e+1 |
| DPA-1 | 2.5e+2 | 2.7e+2 | 7.7e-2 | 1.3e-1 | 8.1e-2 | 9.9e-2 | 7.7e-1 | 6.0e-2 | 8.2e+0 | 7.5e+0 |
| SchNet | 4.1e+2 | 5.0e+2 | 5.1e-2 | 2.4e-1 | 4.5e-2 | 1.5e-1 | 8.0e-1 | 3.8e+0 | 7.2e+1 | 3.8e+2 |
| DimeNet++ | 5.2e+1 | 7.7e+1 | 2.9e-2 | 6.7e-2 | 2.2e-2 | 4.3e-2 | 9.5e-2 | 2.2e+0 | 9.2e-1 | 3.6e+0 |
| GemNet-T | 3.3e+1 | 5.4e+1 | 2.9e-2 | 1.2e-1 | 2.2e-2 | 8.5e-2 | 1.6e-2 | 9.2e-1 | 3.0e-1 | 7.0e+0 |
| GemNet-dT | 4.2e+1 | 6.7e+1 | 2.9e-2 | 6.7e-2 | 2.1e-2 | 8.5e-2 | 6.8e-2 | 1.6e+0 | 6.5e-1 | 2.0e+1 |
| NequIP | 8.3e+1 | 1.2e+2 | 3.3e-2 | 1.3e-1 | 3.1e-2 | 1.5e-1 | 1.4e-1 | 9.0e-1 | 1.9e+0 | 3.5e+0 |
| Allegro | 1.6e+2 | 3.6e+2 | 4.7e-2 | N/A | 4.7e-2 | N/A | 3.0e-1 | 1.7e-1 | 2.6e+0 | 8.3e+0 |
| MACE | 6.6e+1 | 9.2e+1 | 3.2e-2 | N/A | 2.6e-2 | N/A | 5.9e-2 | 1.6e-1 | 5.8e-1 | 8.7e+0 |
| SCN | 4.2e+1 | 6.7e+1 | 2.8e-2 | 6.4e-2 | 2.1e-2 | 5.6e-2 | 4.6e-1 | 8.6e-1 | 5.3e+0 | 5.2e+0 |

Table D.7: Raw scores of models for indicators using HfO dataset and trained with MSE-based loss. Lower values indicate better performance. N/A indicates an interrupted simulation for the dynamic indicator due to abnormal energy changes.

| | EF mtr.$^{ID}$ | EF mtr.$^{OOD}$ | RDF$^{ID}$ | RDF$^{OOD}$ | ADF$^{ID}$ | ADF$^{OOD}$ | $V_0^{ID}$ | $V_0^{OOD}$ | $B_0^{ID}$ | $B_0^{OOD}$ |
|---|---|---|---|---|---|---|---|---|---|---|
| BPNN | 2.2e+2 | 2.3e+2 | 4.5e-2 | 9.1e-2 | 4.5e-2 | 7.6e-2 | 6.3e-1 | 1.2e+0 | 3.6e+0 | 6.3e+0 |
| DPA-1 | 2.3e+2 | 2.6e+2 | 7.1e-2 | 1.3e-1 | 7.4e-2 | 8.9e-2 | 6.3e-1 | 6.4e-1 | 5.9e+0 | 2.8e+0 |
| SchNet | 4.2e+2 | 5.1e+2 | 9.0e-2 | 4.1e-1 | 9.2e-2 | 1.5e-1 | 1.1e+0 | 6.6e+0 | 1.2e+2 | 7.7e+2 |
| DimeNet++ | 4.8e+1 | 7.0e+1 | 3.2e-2 | 1.3e-1 | 2.5e-2 | 1.4e-1 | 6.3e-2 | 1.9e+0 | 1.0e+0 | 3.9e+1 |
| GemNet-T | 4.6e+1 | 6.3e+1 | 3.4e-2 | 6.5e-2 | 2.6e-2 | 5.1e-2 | 4.4e-2 | 9.5e-1 | 2.9e-1 | 1.5e+0 |
| GemNet-dT | 5.7e+1 | 7.5e+1 | 3.2e-2 | 1.6e-1 | 2.5e-2 | 1.3e-1 | 1.7e-1 | 1.1e+0 | 1.5e+0 | 1.3e+1 |
| NequIP | 8.0e+1 | 1.0e+2 | 3.8e-2 | 1.4e-1 | 3.5e-2 | 8.5e-2 | 5.8e-2 | 2.7e-1 | 1.1e+0 | 8.1e-1 |
| Allegro | 1.6e+2 | 2.3e+2 | 4.8e-2 | N/A | 4.7e-2 | N/A | 3.3e-1 | 1.9e-1 | 9.4e-1 | 1.4e+1 |
| MACE | 6.5e+1 | 1.4e+2 | 3.3e-2 | 1.0e-1 | 2.6e-2 | 7.0e-2 | 3.1e-2 | 5.4e-1 | 3.7e-1 | 3.1e+0 |
| SCN | 5.7e+1 | 8.1e+1 | 2.9e-2 | 8.0e-2 | 2.2e-2 | 7.1e-2 | 3.2e-1 | 1.5e+0 | 2.2e+0 | 3.7e+0 |

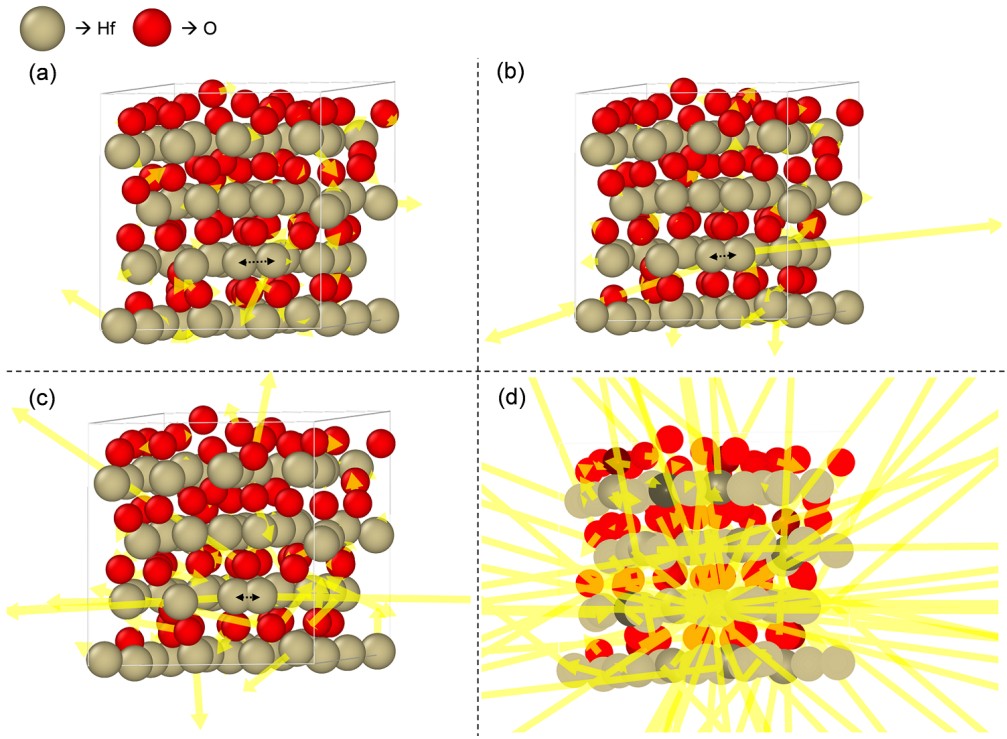

Figure D.8: Hf collision during the evaluation of Allegro in dynamic indicators. (a), (b), (c), and (d) are presented in chronological order, with each atom's relative force magnitude and direction indicated by a yellow arrow.

Table D.8: The maximum threshold ($TH_{max}$) used to transform metrics for the numerical comparison. The transformation rule is as follows: $x' = \frac{TH_{max} - x}{TH_{max} - x_{min}}$, where $x$ and $x'$ represent the original and transformed scores, respectively. $TH_{max}$ is the maximum threshold, and $x_{min}$ represents the minimum value among the model evaluation results, where achieving the minimum signifies the most desirable outcome for each metric.

| Metric | $x_{min}$ | | $TH_{max}$ |
| | HfO | SiN | |
| --- | --- | --- | --- |
| Energy and force errro metric (EF mtr.) | 3.3e+1 | 9.8e+1 | 8.5e+2 |
| Radial distribution function (RDF) | 2.8e-2 | 3.3e-2 | 4.5e-1 |
| Angular distribution function (ADF) | 2.1e-2 | 2.1e-2 | 4.5e-1 |
| Bulk modulus ($B_0$) | 1.9e-1 | 3.9e-1 | 5.0e+1 |
| Equilibrium volume ($V_0$) | 1.6e-2 | 2.5e-2 | 3.0e+0 |

**GemNet-dT** employs direct force prediction, which does not require backpropagation to compute forces from energy. As a result, it provides a high inference speed with low accuracy drop on ID samples, but not on OOD samples.

**NequIP** and **MACE** show high accuracy on both EF metrics and simulation indicators. However, MACE is 1.7x is faster than NequIP with similar accuracy. Compared to GemNet-dT, MACE has equivalent EF accuracy similar prediction results on energy and force with a slightly slower inference speed, but it works better on simulation indicators, and on OOD samples.

**Allegro** has a high inference speed and decent EF accuracy, however, it show perform unstable on MD simulations with OOD samples.

**SCN** achieves high overall accuracy, but has a significantly slow inference speed which is 5x slower than GemNet-T, and 9x slower than MACE.

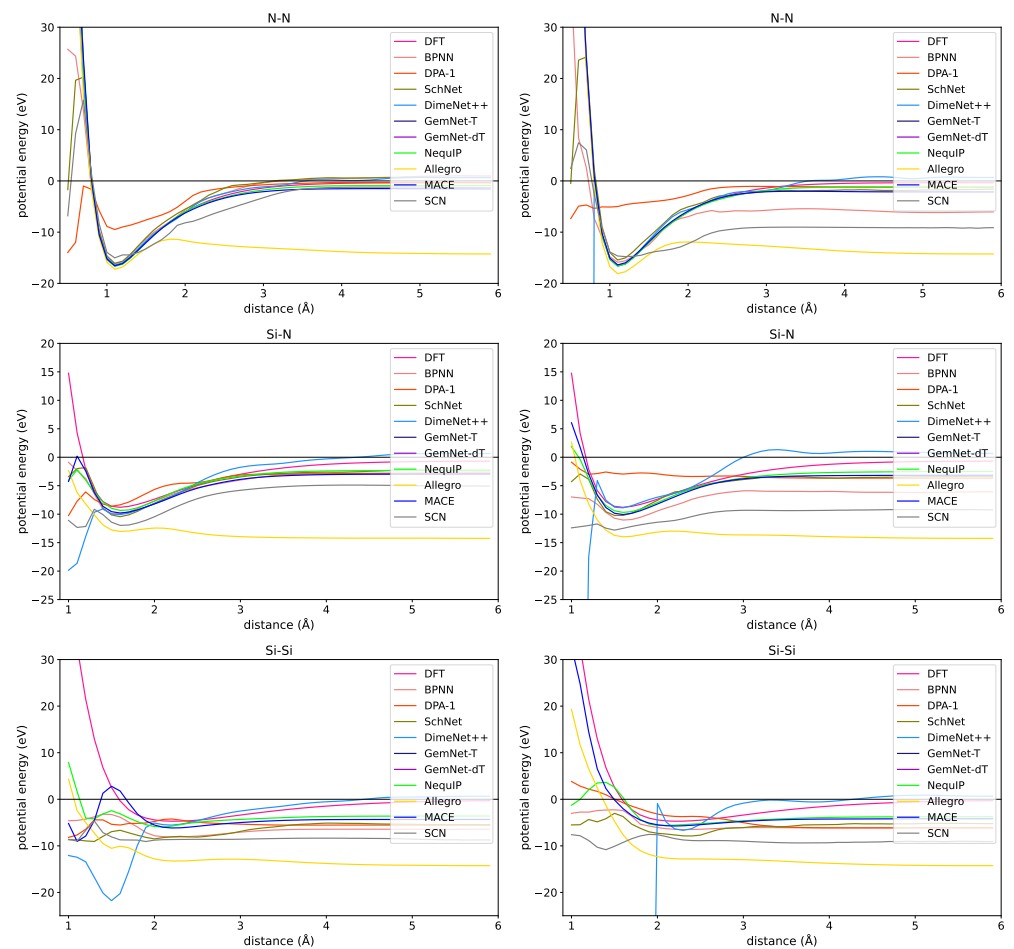

Figure D.9: Comparison of two-body PECs for SiN using MLFF models: models trained with MAE-based loss (left) and MSE-based loss (right).

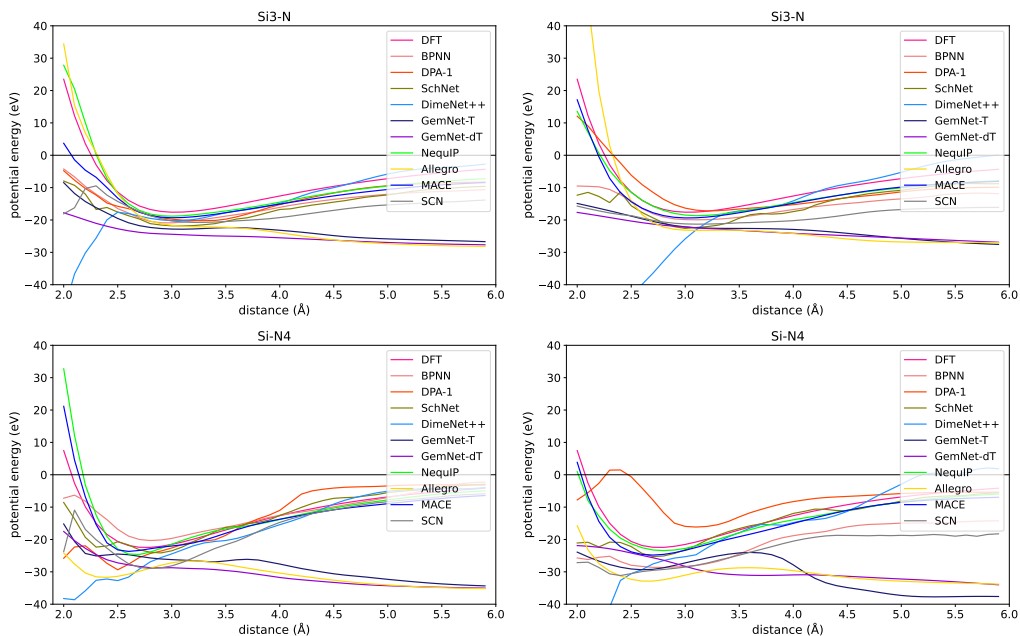

Figure D.10: Comparison of many-body PECs for SiN using MLFF models: models trained with MAE-based loss (left) and MSE-based loss (right).

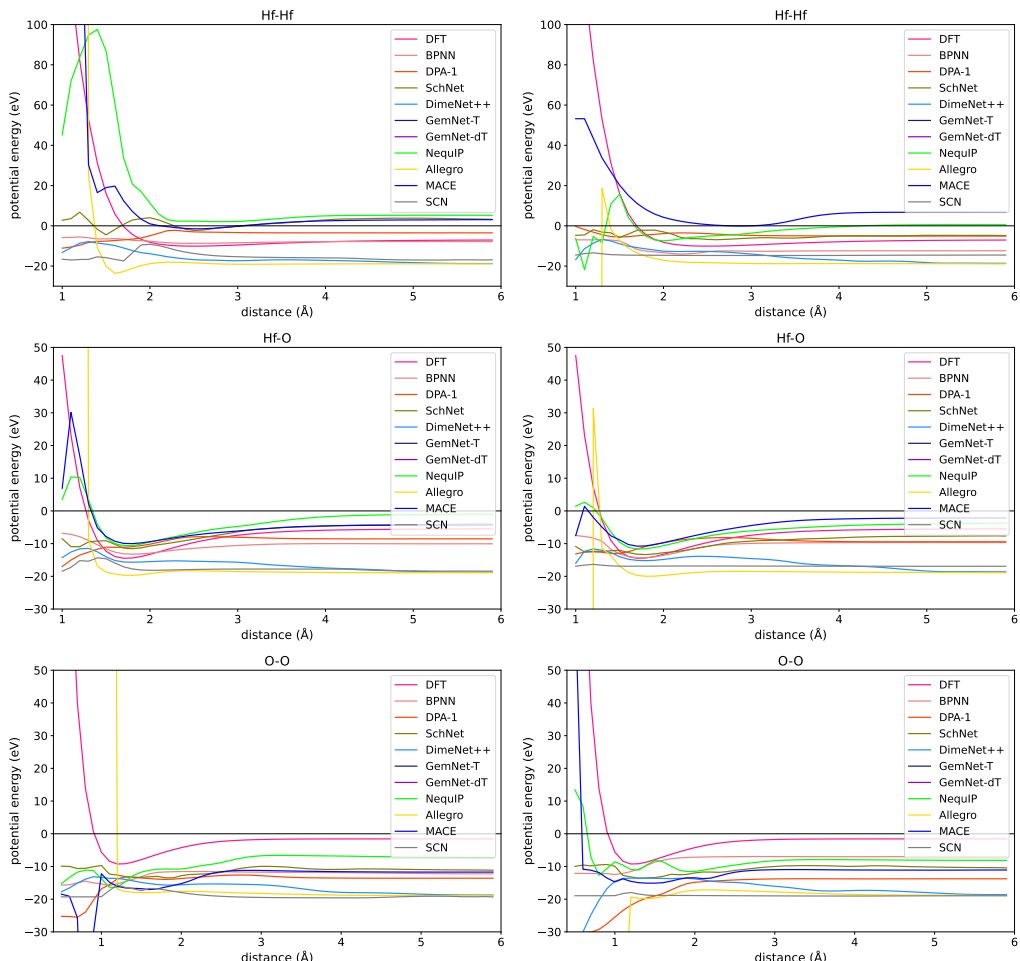

Figure D.11: Comparison of two-body PECs for HfO using MLFF models: models trained with MAE-based loss (left) and MSE-based loss (right).

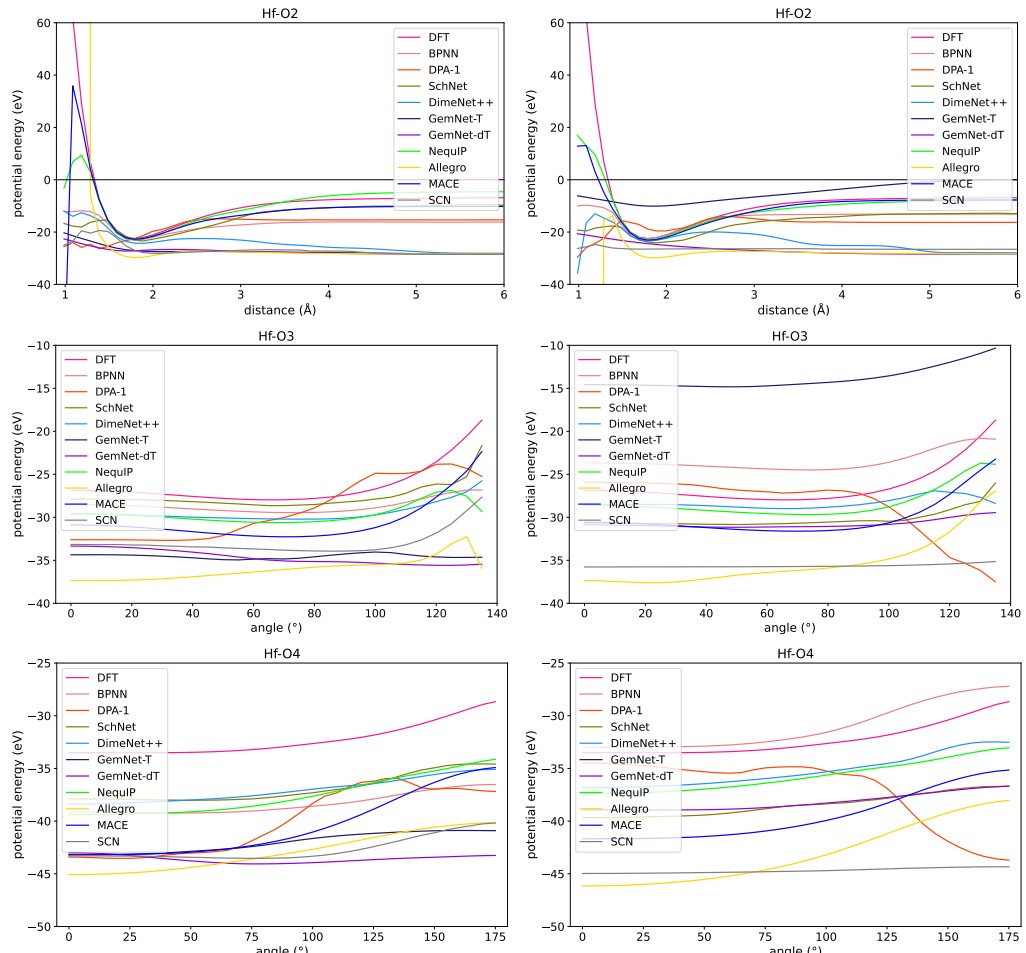

Figure D.12: Comparison of many-body PECs for HfO using MLFF models: models trained with MAE-based loss (left) and MSE-based loss (right).

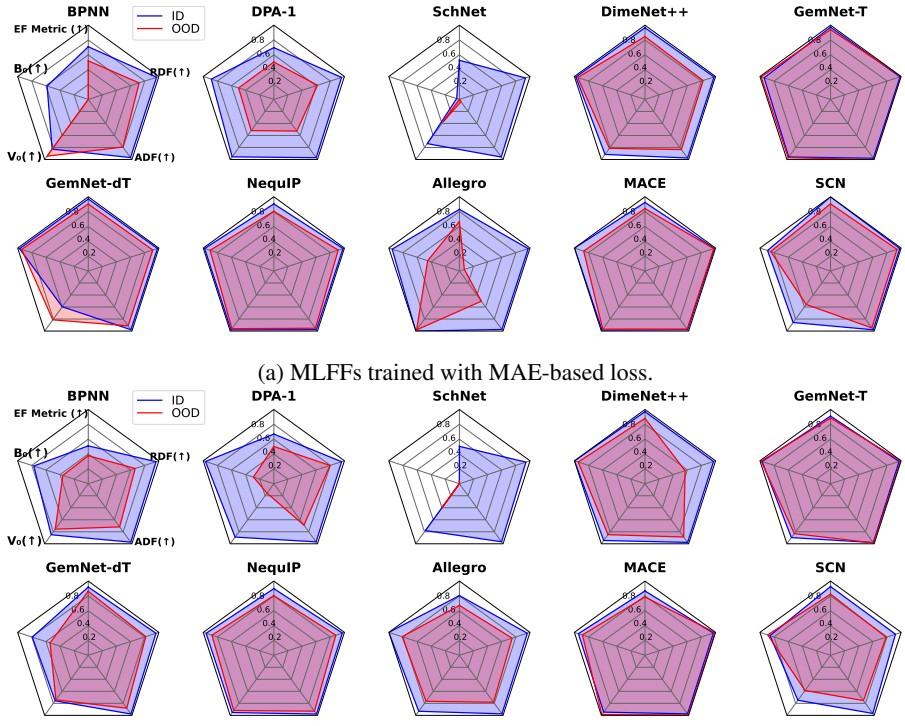

(a) MLFFs trained with MAE-based loss.

(b) MLFFs trained with MSE-based loss.

Figure D.13: Comprehensive comparison of models on the SiN dataset, based on EF metric and simulation metrics. Higher values indicate better performance. The red and blue plots represent the results for ID and OOD, respectively.

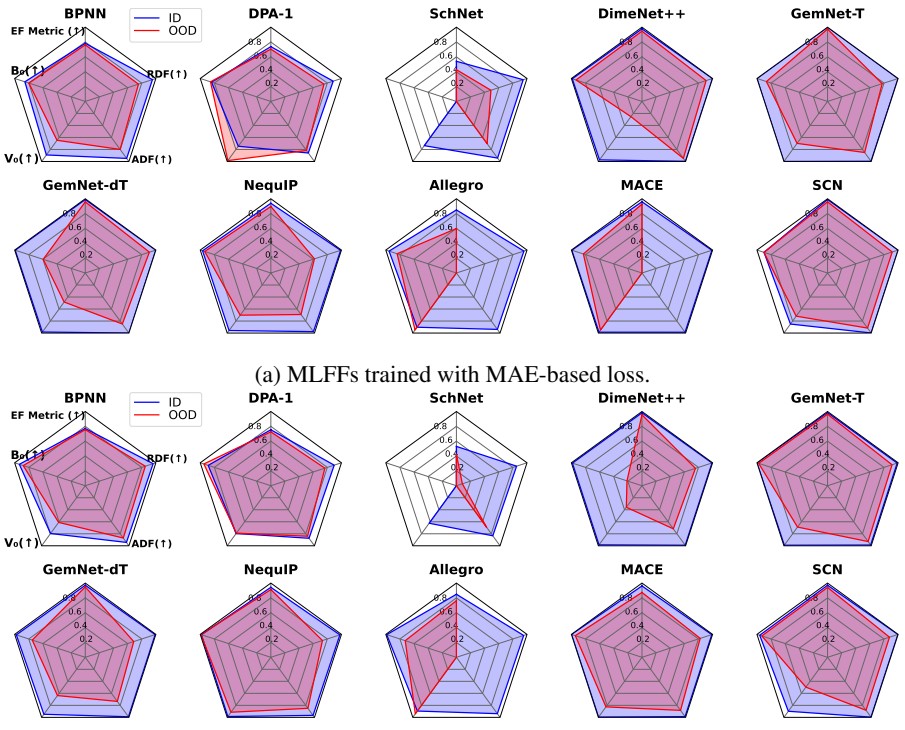

(a) MLFFs trained with MAE-based loss.

(b) MLFFs trained with MSE-based loss.

Figure D.14: Comprehensive comparison of models on the HfO dataset, based on EF metric and simulation metrics. Higher values indicate better performance. The red and blue plots represent the results for ID and OOD, respectively.

