# OpenReview forum: "Benchmark of Machine Learning Force Fields for Semiconductor Simulations: Datasets, Metrics, and Comparative Analysis"
_NeurIPS.cc/2023/Track/Datasets_and_Benchmarks — NeurIPS 2023 Datasets and Benchmarks Poster_

### Official Review · Reviewer_6cbt · 2023-07-21
**A useful dataset for benchmarking machine learning force fields**

**Rating:** 6
**Confidence:** 5
**Correctness:** Both the dataset construction and eva…

**Strengths:**

- The paper generates new benchmark datasets using DFT calculations.
- The paper introduces OOD test data based on amorphous structures, which is beneficial to evaluate the generalizability of MLFFs.
- The paper introduces simulation and static metrics. These metrics are useful for evaluating the MLFF’s performance in a realistic scientific setting.
- Both descriptor-based and GNN-based MLFF are selected to benchmark, offering a comparison with non-deep-learning methods.


**Additional Feedback:**

No additional feedback.

**Clarity:**

The paper is well-written. Lot of information referenced in the main text are in the supplementary information, which makes it hard to read occationally.


**Documentation:**

- The paper clearly documented the details for data generation, including DFT software, settings, etc.
- The linked URL only contains raw data files. It does not contain instructions for intended use
- The linked URL does not contain the train/validation/test split used in the paper.
- Detailed instructions for reproducing the benchmark can be found in the supplementary information.


**Ethics:**

There is not ethics concerns.

**Limitations:**

The authors didn’t discuss limitations and negative social impact in their paper. I suggest the authors include a paragraph to discuss the limitations of their dataset and evaluation approaches and potential direction to improve it in the future.

**Opportunities For Improvement:**

- The dataset only includes 2 materials, HfO and SiN and is not very diverse. It would also be beneficial if more complex materials can be included.
- How do the authors split the train, validation, and test data?
- It is nice to add a data scaling plot to show the change of performance with respect to the size of training data


**Relation To Prior Work:**

There has been extenstive MLFF benchmarks developed recently. The authors didn't adequatly discuss the difference of their work with priori benchmarks. The authors should considering discuss their work's difference with e.g., [1], [2], and others.

[1] Forces are not Enough: Benchmark and Critical Evaluation for Machine Learning Force Fields with Molecular Simulations

[2] The Open Catalyst 2022 (OC22) Dataset and Challenges for Oxide Electrocatalysts

**Summary And Contributions:**

This paper creates two new benchmark datasets for training MLFF on semiconductor solid materials – HfO and SiN. The benchmark features OOD test sets and additional simulation/static metrics to evaluate the MLFFs. It benchmarked 10 models including descriptor-based and GNN-based methods.

---

> ### Author Response · Authors · 2023-08-18
> **Authors' Response to Reviewer 6cbt (Part I)**
>
> We sincerely appreciate the invaluable feedback and insights you've provided on our manuscript.  We have taken your comments very seriously and have made substantial edits to our manuscript to address the points raised.
>
> > The dataset only includes 2 materials, HfO and SiN and is not very diverse. It would also be beneficial if more complex materials can be included.
> - We'd like to clarify that we prioritized the release of datasets focusing on SiN and HfO materials, which have been receiving significant attention in the realm of next-generation semiconductor device design. Notably, SiN stands out as a potential next-generation storage medium, while HfO is being recognized for its critical role as a high-k dielectric and a vital ferroelectric material. Active computational simulation research is being carried out on these materials. Our decision to debut with these materials was strategically made, hoping to foster increased interest in MLFF within the research community.
>
>   We understand the concerns about the limited variety of our current datasets. Although it's challenging during this rebuttal period to address this in its entirety, we are eager to inform that datasets on the Hf-Zr-O and the Hf-Si-O ternary systems are nearing completion and will be introduced to the community in the near future. Furthermore, we have plans in motion to sequentially unveil diverse semiconductor datasets, which include ternary systems or higher, consisting of atoms such as Si, Ge, C, N, O, H, Hf, Zr, B, P, Ga, and As.
>
>   It's worth noting that while we aim for diversity, we must strike a balance to maintain the quality of our datasets. Based on our internal tests, we have found that datasets that span too many elements can sometimes degrade model stability and their adaptability to varied simulations. Such comprehensive datasets often capture only a narrow atomic environment for each material, thus limiting their utility for meaningful research outcomes.
>
>   We recognize that in the semiconductor field, data sharing is often a challenge for many groups. However, we firmly believe that our initiative in releasing data will be pivotal for the progress of MLFF models and simulation studies. We remain dedicated to consistently introducing datasets for a range of materials in the coming times.
>
> > How do the authors split the train, validation, and test data?
> - It appears that there may have been some ambiguity in our initial documentation. In the link initially submitted, there are four distinct tarballs. Among these, the specific train:validation:test splits and the OOD sets used in our study are thoroughly shared within the **HfO.tar** and **SiN.tar** files. Additionally, **HfO_raw.tar** and **SiN_raw.tar** are the raw databases which serve as the foundational datasets from which all subsequent splits were derived. We've also ensured that every file is standardized in the extended-xyz format for consistency.
>
>   To avoid any confusion, we have taken immediate steps to enhance clarity and accessibility. Our README file in the submitted code material has been updated to include comprehensive descriptions of the four distinct tarballs available in the submitted link. To provide clarity on our data processing, we have also shared the Python scripts used for sampling and splitting, which include the information of sampling intervals.
>
>   Regarding the SiN dataset, we extracted MD snapshots at 9 fs intervals from each ground truth simulation scenario. For the HfO dataset, while the pre-melting and melting phases used a 9 fs interval, the subsequent quenching and annealing phases were sampled at 12 fs intervals. To achieve a diverse representation, these snapshots were then randomly divided into training, validation, and test sets in an 8:1:1 ratio. This approach was adopted to ensure the inclusion of diverse atomic structures and avoid redundancy in our dataset.
>
>   Beyond the specific splitted datasets, we released the raw datasets, anticipating it could provide broader avenues for research in the MLFF domain, as discussed in the **Appendix A.1** (Guidance). Given the early stage of discussions on data splits and efficiency in the field, our aim is to encourage diverse research endeavors by providing more comprehensive access.

---

> ### Author Response · Authors · 2023-08-18
> **Authors' Response to Reviewer 6cbt (Part II)**
>
> > It is nice to add a data scaling plot to show the change of performance with respect to the size of training data.
>
> - This suggestion would enhance the quality of our paper by providing beneficial benchmark results that may be helpful to make a training strategy for MLFF researchers. To investigate data scaling effect, we follow the experimental settings of previous works (NequIP, MACE), where the ReduceLROnPlateu scheduler is used to train MLFF models. Even though the training steps cannot be fixed due to this scheduler unlike our setting, we anticipate that the training might be stopped at similar update steps. Thus, in the data scaling experiments, MLFF models are trained by the fixed number of steps for model update; the setting is referred to as the equal budget setting. We randomly sample 20%, 40%, 60%, and 80% of data from the HfO training set (5.6k, 11.2k, 16.8k, and 22.4k), and trained models by 1000, 500, 334, and 250 epochs, correspondingly. We select six models (BPNN, GemNet-T, GemNet-dT, NequIP, Allegro, and MACE) and obtain their energy and force errors using the HfO test and OOD sets. Except the data size and epochs, the training recipe used in the MAE-based training experiments of Figure 1 is maintained.
>
>   The results of the EF metric are added in the **Appendix C.5** (Table C.5 and Figure C.4) and also included in this response below as a table. For all models, as more training data are used, the training error increases while the test error decreases, where the gap between these two errors also decreases. The OOD error decreases in most cases. Therefore, our data scaling results suggest that training with fewer epochs and more data can be helpful to improve the generalization performance of MLFF models. Meanwhile, the difference between EF metrics using 80% and 100% training set is marginal, meaning that the performance improvement that can be gained by using more data seems to be saturated. Such observation implies that it is reasonable that our training set sampled from the raw dataset is sufficient for semiconductor MLFF benchmark.
>
>   We can opt for another data scaling experimental setup where the identical training epochs are used to train MLFF models; the setting is referred to as the equal epoch setting. In the equal epoch setting, as the training set size is reduced, the number of model update steps is also reduced. Considering that accuracy drop is clearly observed when using less data at the equal budget setting, we omit the experiments of the equal epoch setting because further accuracy drop is intuitively expected.
>
> - | Model | num train | Train | Test | OOD | \| | Model | num train | Train | Test | OOD |
> |-|-|-:|-:|-:|-|-|-|-:|-:|-:|
> | BPNN | 5.6k | 190.1 | 231.4 | 260.6 | \| |NequIP | 5.6k  | 72.5 | 99.4 | 174.6 |
> | | 11.2k | 200.9 | 216.4 | 239.1 | \| | | 11.2k | 76.6 | 88.1 | 148.5 |
> | | 16.8k | 204.1 | 212.9 | 232.4 | \| | | 16.8k | 78.1 | 84.2 | 130.0 |
> | | 22.4k | 205.8 | 210.8 | 230.8 | \| | | 22.4k | 79.0 | 82.5 | 119.1 |
> | | 28.0k | 206.4 | 210.2 | 227.4 | \| | | 28.0k | 79.8 | 82.6 | 117.8 |
> |||||||||
> | GemNet-T | 5.6k | 19.8 | 48.9 | 75.2 | \| | Allegro | 5.6k | 144.7 | 174.6 | 1923.8 |
> | | 11.2k | 23.6 | 39.3 | 64.7 | \| | | 11.2k | 148.9 | 163.8 | 658.3 |
> | | 16.8k | 25.1 | 35.2 | 59.2 | \| | | 16.8k | 150.8 | 157.5 | 365.8 |
> | | 22.4k | 26.0 | 33.3 | 54.1 | \| | | 22.4k | 152.0 | 156.4 | 415.3 |
> | | 28.0k | 27.3 | 33.0 | 54.3 | \| | | 28.0k | 152.7 | 156.0 | 360.5 |
> |||||||||
> | GemNet-dT | 5.6k | 25.1 | 62.4 | 91.8 | \| | MACE | 5.6k | 61.7 | 71.0 | 221.6 |
> | | 11.2k | 31.7 | 49.0 | 79.2 | \| | | 11.2k | 64.3 | 68.3 | 180.4 |
> | | 16.8k | 34.4 | 44.4 | 73.0 | \| | | 16.8k | 65.1 | 67.6 | 126.0 |
> | | 22.4k | 35.3 | 42.1 | 67.0 | \| | | 22.4k | 66.4 | 67.3 | 114.6 |
> | | 28.0k | 36.3 | 41.9 | 67.2 | \| | | 28.0k | 66.1 | 66.3 | 92.2 |

---

> ### Author Response · Authors · 2023-08-18
> **Authors' Response to Reviewer 6cbt (Part III)**
>
> > I suggest the authors include a paragraph to discuss the limitations of their dataset and evaluation approaches and potential direction to improve it in the future.
>
> - We have reinforced and relocated the paragraph "Limitation" and "Future plan" from Appendix to the main paper (**Section 6**). Other than our dataset expansion plan to Hf-Si-O and Hf-Zr-O systems, we are planning to adopt efficient training schemes such as transfer learning, to reduce model training cost for growing datasets. We also plan to subjoin metrics and loss factors (e.g., stress) to evaluate compatibility with various simulation conditions (e.g., NPT), and reflect indicators during training to provide better prediction performance. Moreover, we will extend our benchmark to include compression and parallelization performance of the MLFF models to assess suitability for large-scale simulations.
>
> > The authors didn't adequatly discuss the difference of their work with priori benchmarks. The authors should considering discuss their work's difference with e.g., [1], [2], and others.
> - From this feedback, we recognized the importance of distinguishing our work from existing benchmarks in the MLFF domain, and thus we have made substantial edits to our manuscript to address the points raised. In the revised manuscript, specifically in Section 2 under 'MLFF Benchmark' (**line #76~88**), we have outlined the distinctions between our dataset and other benchmarks, such as OC20 and OC22. We highlighted the importance of our dataset in semiconductor material research and addressed challenges that aren't covered by current datasets. While we reference the "Forces are not Enough" paper to highlight existing views on MLFF evaluations, we also emphasized the need for a tailored benchmark for semiconductor research and pointed out how ours stands apart. We believe these edits make our contributions clearer and address the initial concerns of the reviewers.
>
> We are grateful for this opportunity to emphasize the unique aspects of our research. We are confident that these changes help to clarify our novel contributions. Your feedback has been pivotal in enhancing our manuscript, and we deeply appreciate it.

---

> ### Author Response · Authors · 2023-08-28
> **Seeking the Reviewer's Insights on Our Updated Manuscript**
>
> Dear Reviewer 6cbt,
>
> We sincerely value the insightful feedback you provided on our manuscript. As there is approximately one and a half days left in the author-reviewer interaction period, we're reaching out to kindly inquire if our previous responses met your expectations and to ask about any further comments or concerns you might have. We genuinely hope to ensure that all our revisions and responses align with your feedback, and we are committed to carefully considering and addressing any further suggestions you may provide.
>
> We deeply appreciate your time and dedication to this review process.
>
> Respectfully,
> Authors

---

> > ### Comment · Reviewer_6cbt · 2023-08-28
> > **Thank you for detailed reponse!**
> >
> > I thank the authors for their detailed response to my comments. I've read the comments and comments from other reviewers. I am satisfied with the authors replies.

---

### Official Review · Reviewer_dwvr · 2023-07-22
**Discussion of this paper**

**Rating:** 6
**Confidence:** 1
**Clarity:** Yes, it is well-written.

**Strengths:**

1. The dataset and benchmark track in this paper offer significant value due to the utilization of snapshots generated by density functional theory (DFT) during relaxation. While this approach ensures high-quality data, it also comes with a trade-off as it involves a time-consuming and computationally expensive generation process. Despite these challenges, the adoption of DFT snapshots enhances the dataset's credibility and relevance, making it a valuable resource for researchers in the field of machine learning force fields (MLFF).

2. The paper stands out for its exceptional clarity and coherence in presenting complex concepts. The authors have demonstrated their adeptness in conveying core ideas and main contributions with remarkable ease. Through concise and well-structured writing, readers can effortlessly grasp the essential aspects of the research, facilitating a deeper understanding of the proposed datasets, benchmark methods, and their implications for the machine learning community.






**Additional Feedback:**

None.

**Correctness:**

I think the conclusions are all drawn from the extensive empirical studies, which are conducted in a technically sound way and can be reproduced.

**Documentation:**

I think there is sufficient material for ML community to use this dataset and reproduce benchmarked models.

**Ethics:**

None.

**Limitations:**

No negative societal impact.

**Opportunities For Improvement:**

It would be better to provide a in-depth empirical analysis of the advantages and disadvantages of each model, and summary comprehensive future directions based on this analysis. In this way, this benchmark tend to motivate more and more MLFF research works.

**Relation To Prior Work:**

It seems that this is the first datasets about semiconductor's FF predictions.

**Summary And Contributions:**

In this paper, the authors address the existing gap in the availability of a dedicated benchmark for machine learning force field (MLFF) in the realm of computational material research, catering to the needs of machine learning researchers. To fill this void, they present two interconnected datasets specifically designed for MLFF, against which they evaluate the performance of ten relevant methods using six key metrics. The datasets are constructed using density functional theory (DFT), requiring substantial computational resources in the form of extensive GPU hours. By introducing these benchmark datasets and evaluating various methods, the authors aim to provide valuable support to the machine learning community, fostering accelerated progress in the field of MLFF research.

---

> ### Author Response · Authors · 2023-08-18
> **Authors' Response to Reviewer dwvr**
>
> We are grateful for the feedback and for taking the time to review our manuscript. We have incorporated your recommendations for further improvement.
>
> > It would be better to provide a in-depth empirical analysis of the advantages and disadvantages of each model.
>
> - We included empirical analysis with advantages and disadvantages of each model in the appendix (**Section D.5**), as the following.
>
>   **BPNN** requires hand-crafted features but has simplest overall structure, which makes the model relatively fast and less accurate than recent SOTA models.
>
>   **DPA-1** has relatively low accuracy on the EF metric, but shows better performance on simulation indicators.
>
>   **SchNet** lies on the pareto-frontier as the fastest model, but the speed gain may not be enough to compensate overall accuracy drop compared to recent fast models such as Allegro or GemNet-dT.
>
>   **DimeNet++** and **GemNet-T** are models with similar base structure. They show a similar overall tendency: generally high accuracy but less accurate on the V0 indicator, and having relatively slow inference speed. GemNet-T, which lies on the most accurate side of pareto-frontier, seems slightly better predictive performance than DimeNet++.
>
>   **GemNet-dT** employs direct force prediction, which does not require backpropagation to compute forces from energy. As a result, it provides a high inference speed with low accuracy drop on ID samples, but not on OOD samples.
>
>   **NequIP** and **MACE** show high accuracy on both EF metrics and simulation indicators. However, MACE is 1.7x is faster than NequIP with similar accuracy. Compared to GemNet-dT, MACE has similar prediction results on energy and force with a slightly slower inference speed, but it works better on simulation indicators, and on OOD samples.
>
>   **Allegro** has a high inference speed and decent EF accuracy, however, it perform unstable on MD simulations with OOD samples.
>
>   **SCN** achieves high overall accuracy, but has a significantly slow inference speed which is 5x slower than GemNet-T, and 9x slower than MACE.
>
> > It would be better to provide summary comprehensive future directions based on this analysis.
>
> - We also reinforced and relocated the paragraph to suggest future directions in the main paper (**Section 6**). From the model analysis we have found that inference speed of a MLFF model does not have a clear tendency with the model size. However, since the major purpose of MLFF on semiconductors is to enable large scale simulation that traditional ab-initio MD cannot perform, more analysis for model inference speed is required. Thus, we will extend our benchmark to include compression and parallelization performance of the models. Furthermore, our future direction includes expanding datasets and reflecting extra components or indicators during training and evaluation, to make compatible with various simulation conditions.
>
> We sincerely hope that our revisions offer a more comprehensive perspectives on the research. We remain deeply appreciative of your feedback and are open to any further insights or suggestions you might have.

---

> > ### Comment · Reviewer_dwvr · 2023-08-23
> > **Discussions**
> >
> > Thanks for your response! Most of my concerns are resolved now. I will keep the score to support this paper.

---

### Official Review · Reviewer_PKAy · 2023-07-26
**This paper provides benchmarks/datasets for MLFF simulation. Though it looks complete as a dataset introduction work, the paper requires significant domain knowledge for readers to follow.**

**Rating:** 5
**Confidence:** 3
**Correctness:** Reasonably correctness.
**Clarity:** Medium.

**Strengths:**

The dataset is reasonably large and well-constructed for ML application.

**Additional Feedback:**

Please refer to above comments.

**Documentation:**

Documentation is quite limited.

**Ethics:**

Not found.

**Limitations:**

Similar to above comments, the major limitation of the paper is the writting is too domain-specific. Need to be expressed clearly to non-domain readers as a general ML benchmarking or dataset paper. Also the problem definition and data format needs to be clearly introduced.

**Opportunities For Improvement:**

Though the reviewer is not an expert in Semiconductor material film simulation. I would recommend to make this paper general to non-domain readers. i.e., include more details of the problem description and the dataset in mathematical or machine learning language.
1) What are the input and output of the problem, and what is the data format (tensor, vector or some graph representations).
2) What are the evaluation metrics, equations to compute them. Are they compared to the real physical simulation?
3) Authors also proposed 5 additional indicators. How are these indicators related to the applicability of the MLFF model? What are the value ranges of these indicators? Smaller is better or larger is better?
4) Besides the details, there should also be discussion of existing benchmarks/datasets, how the proposed dataset is better than previous one?

**Relation To Prior Work:**

Needs improvements.

**Summary And Contributions:**

This paper provides benchmarks/datasets for MLFF simulation. The major contributions include:
1) SiN and HfO dataetes for MLFF simulation.
2) Experiments on SOTA MLFF models and provide evaluation results.
3) Provide additional simulation indicators as evaluation metric.

---

> ### Author Response · Authors · 2023-08-18
> **Authors' Response to Reviewer PKAy**
>
> Thank you for your thoughtful feedback and the dedicated time you've allocated to reviewing our manuscript. We deeply value your comments and consider them instrumental in enhancing our work.
>
> > What are the input and output of the problem, and what is the data format (tensor, vector or some graph representations). What are the evaluation metrics, equations to compute them. Are they compared to the real physical simulation?
>
> - We apologize for any confusion in understanding MLFFs. The description in Section 2 (**line #61~63**) was briefly written due to the page limitation, and thus may not be sufficient to fully understand the problem setup from a machine learning researcher's perspective. To help readers understand MLFF more clearly, we have added an illustration for the MLFF workflow at the **Appendix B**.
>
>   The input data has a form of 3D point cloud data, which consists of coordinates $r_i = (x_i, y_i, z_i)∈R^3$, and atom numbers $Z_i∈Z$ for $1 ≤ i ≤ n$, where n represents the number of atoms. Occasionally, input data is also considered as a graph data, but to be precise, it is a preprocessed data with a 3D graph form, by constructing edges between points within a cutoff radius. This conversion is not mandatory, but commonly employed by GNN based models. The model output consists of the (potential) energy $E∈R$ and force $F_i = (F^x_i, F^y_i, F^z_i) ∈R^3$ for $1 ≤ i ≤ n$. As a language of the physical simulation, the energy of the system indicates its potential stability. The forces, which are induced by the energy, are integrated over time to determine atomic trajectories, guiding the movement and interactions of atoms in the system.
>
>   The EF metric suggested in the paper combines the RMSE of per-atom energy and the Axis-wise RMSE of force, both of which are presented in **Table C.1**, to provide a comprehensive assessment of the accuracy of predicted energies and forces by the MLFF models. A precise prediction of both energy and force is crucial because any inaccurate prediction can lead to incorrect atom movements and, subsequently, unreliable simulation outcomes.
>
> > Authors also proposed 5 additional indicators. How are these indicators related to the applicability of the MLFF model? What are the value ranges of these indicators? Smaller is better or larger is better?
>
> - In response to your concerns, we have included comprehensive sections in the manuscript (**Section D.1.1, D.2.1, and D.3.1**) that elucidates the significance and relevance of our proposed indicators. We provide a more comprehensive explanation of each metric, including RDF, ADF, Equation of State metrics, and PECs, specifically tailored for individuals without a background in physics expertise. These metrics are now analyzed in the context of real-world simulations and for the applicability of model evaluations. Through these expanded sections, we aim to clarify the role of each indicator in assessing the performance of our MLFF models.
>
>   We've also included clear explanations that dictate the optimal performance for these metrics, indicating whether a smaller or larger value is preferable. For enhanced clarity, we have incorporated arrows, and corresponding explanations in all figures and tables that illustrate inter-model comparisons of indicators. Furthermore, all metrics were formulated based on the difference error from the real simulation, DFT golden standard, with smaller values indicating better performance. Except for cases like V0 where experimental evidence exists (please refer to **Section D.2.1**), there is no predefined threshold for how small the values should be maintained even in the simulation domain. However, we have comparatively evaluated the state-of-the-art models in a relative context and have explicitly specified these values, making them applicable as reference points for comparison. The obtained results for each metric from each model are delineated in **Tables D.4 to D.7**, and the best results are reported in **Table D.8**. We value the readability suggestions and are confident they'll enhance our paper's quality.
>
> > Besides the details, there should also be discussion of existing benchmarks/datasets, how the proposed dataset is better than previous one?
>
> - Recognizing the pointed lack of discussion on existing benchmarks/datasets and the need for a comparison with previous datasets, we have revised Section 2 of the paper, specifically the MLFF Benchmark section (**line #76~88**). This revision encompasses an expanded discussion of recent MLFF datasets and benchmarks, and further articulates the significance of our benchmark and dataset. Moreover, discussions on existing large-scale structural databases have been added to the supplementary **Section A.3**. We hope that these modifications sufficiently address the provided comments.
>
> Once again, we deeply appreciate the constructive feedback, which has undoubtedly contributed to enhancing the quality of the paper.

---

> > ### Comment · Reviewer_PKAy · 2023-08-22
> >
> > The reviewer appreciates the efforts in elaborating on the problem and the dataset. This benchmark looks interesting to the field of AI for science. Regarding the revision, I would like to change my rating to 5.
> >
> > 1. I would also suggest authors make some discussion on the importance of AI/ML to solve such problems. Are SOTA AI models already doing great on the force prediction that can be used for production analysis? If not, how far is it? What are the impacts of the gap between ML model and the simulation results? This is somehow related to the motivation of developing AI benchmarks.
> >
> > 2. Another major concern as a high-quality benchmark and dataset: there should be clear and comprehensive documentation on the dataset and usage instructions.
> >
> > 3. Authors also did some scaling studies as shown in supplementary docs. A minor comment is the scaling of the dataset is usually happening in the order of magnitudes (1K 10K 100K etc.) in order to show more meaningful results.

---

> > > ### Author Response · Authors · 2023-08-25
> > > **Authors' Additional Response to Reviewer PKAy (Part I)**
> > >
> > > Again, we appreciate for your additional feedback. Thankfully, your immediate comments to our first response can help to improve our paper further.
> > >
> > >
> > > > I would also suggest authors make some discussion on the importance of AI/ML to solve such problems. Are SOTA AI models already doing great on the force prediction that can be used for production analysis? If not, how far is it? What are the impacts of the gap between ML model and the simulation results? This is somehow related to the motivation of developing AI benchmarks.
> > >
> > >
> > > - Although SOTA GNN-based MLFF models are good at predicting energy and force for each snapshot, they have only been adopted to limited MD simulations. For semiconductor materials on which we focus, MLFFs are still underutilized. Reasons for the underutilization may include the insufficiency of semiconductor material datasets, model inference speed within MD simulations, and simulation stability. In this paper, we deal with these issues from the perspective of researchers using simulations.
> > >
> > >   Firstly, we have released datasets with diverse simulations, and we hope this will spur further development of MLFF models for semiconductor materials.
> > >
> > >   In terms of stability, researchers may hesitate to employ the GNN-based models because high energy and force prediction power does not perfectly guarantee accurate and stable simulations. To resolve this issue, we proposed five indicators that can reflect model stability and prediction performance in simulations. In addition, through the proposed indicators, we tried to present the comparative advantages and characteristics of the models, aiming to provide valuable insights for making informed decisions in model selection.
> > >
> > >   Regarding inference speed, we listed inference time of MLFF-based models with some architectural variations (Figure 3) and their tradeoff between speed and accuracy. However, for large-scale MD simulations, multi-GPU parallelization is essential to employ the SOTA GNN-based MLFF models. Due to the message passing in GNNs, communication between GPUs is required at every interaction layer. Unfortunately, effective and efficient solutions have been rarely studied. Hence, classical shallow models (e.g., MLP-based models using descriptors calculated from 3D position data by theory-based feature engineering) are widely adopted in MLFF-based simulations despite their less accuracy than GNN-based models. But, we believe that lightweight models which can be used for large-scale simulations will be developed, and our datasets can be one of the key datasets to foster such research direction.
> > >
> > >
> > > > Another major concern as a high-quality benchmark and dataset: there should be clear and comprehensive documentation on the dataset and usage instructions.
> > >
> > > - Based on the feedback, we have incorporated detailed usage instructions into **Appendix A.5**. This section clarifies the dataset's structure and provides guidance on its proper interpretation and utilization. Additional guidance for data manipulation is available in **Appendix A.1**, where we outline various approaches to designing experiments with our dataset. **Appendix A.2** further delves into the dataset's characteristics, as illustrated in **Figures A.2** through **A.5**. Our aim has been to offer insights with greater depth and clarity than what is typically seen in current MLFF dataset publications. Moreover, we have added further details about the shared dataset in the README file of our recent code submission, aiming to enhance comprehension. We genuinely value your feedback and are committed to ensuring our dataset is user-friendly, catering not only to MLFF experts but also to the wider ML community. We'll continue to explore ways to improve our documentation.

---

> > > ### Author Response · Authors · 2023-08-25
> > > **Authors' Additional Response to Reviewer PKAy (Part II)**
> > >
> > > > A minor comment is the scaling of the dataset is usually happening in the order of magnitudes (1K 10K 100K etc.) in order to show more meaningful results.
> > >
> > > - We appreciate your feedback on the data scaling experiments presented in our initial author response.
> > >
> > >   We didn't initially perform a log scaling experiment because we first noticed a considerable accuracy drop when training with only 20% (5.6k) of our original training set. Intuitively, there was no necessity to train with even smaller datasets like 1% or 10%. Hence, we opted for a linear scaling experiment setup. In addition, such scaling setup can be seen in NequIP paper (Nature Communications, 2022).
> > >
> > >   Nevertheless, your suggestion to perform a log scaling experiment is reasonable, and we decided to try to resolve your comment as best as we can. The setup of our second data scaling experiments using the HfO dataset is described as follows. Note that, as described in the paper (**Section 3.1**), our original training/validation/test sets were derived by sampling from the HfO raw dataset and subsequently partitioning into 8:1:1. The new training set used in the second data scaling experiments includes all snapshots from the raw dataset except for those included in the original valididation and test sets, resulting in a total of 152,940 snapshots, referred to as the HfO-all set. From the HfO-all set, we sample 50% (76.5k) and 1% (1.5k), and train models for 73 epochs and 3655 epochs, respectively. For a quick response, we compared the results of additional experiments with our initial data scaling experiment and prioritized testing GemNet-dT and MACE before other models.
> > >
> > >   We append the new results to the initial data scaling results. As seen in the table below, both models trained with 1% of HfO-all set experience the siginificant increase in the test errors while the training errors are much reduced, meaning that overfitting occurs. From the results using more training data than our original training set (i.e., 50% and 100% in the table), we observed no significant performance enhancement, indicating it is not necessary to include all snapshots of the DFT-based simulation into the training set in the MLFF task. It may be insightful to investigate the reason why the composition method with sampling by an interval as in ours can be effective.
> > >
> > >
> > > - | Model | num train (ratio to HfO-all) | Train | Test | OOD |
> > > |-|-|-:|-:|-:|
> > > | GemNet-dT | 1.5k (1%) | 8.5 | 123.5 | 186.4 |
> > > | | 2.8k (1.8%)   | 17.5 | 82.9 | 119.4 |
> > > | | 5.6k (3.6%)   | 25.1 | 62.4 | 91.8 |
> > > | | 11.2k (7.3%)  | 31.7 | 49.0 | 79.2 |
> > > | | 16.8k (11.0%) | 34.4 | 44.4 | 73.0 |
> > > | | 22.4k (14.7%) | 35.3 | 42.1 | 67.0 |
> > > | | 28.0k (18.3%) | 36.3 | 41.9 | 67.2 |
> > > | | 76.5k (50.0%) | 33.5 | 39.8 | 66.0 |
> > > | | 152.9k (100%) | 35.7 | 41.2 | 68.9  |
> > > ||||||
> > > | MACE | 1.5k (1%) | 42.2 | 230.5 | 184.1 |
> > > | | 2.8k (1.8%)   | 56.9 | 77.8 | 410.2 |
> > > | | 5.6k (3.6%)   | 61.7 | 71.0 | 221.6 |
> > > | | 11.2k (7.3%)  | 64.3 | 68.3 | 180.4 |
> > > | | 16.8k (11.0%) | 65.1 | 67.6 | 126.0 |
> > > | | 22.4k (14.7%) | 66.4 | 67.3 | 114.6 |
> > > | | 28.0k (18.3%) | 66.1 | 66.3 | 92.2  |
> > > | | 76.5k (50.0%) | 58.1 | 67.1 | 68.4 |
> > > | | 152.9k (100%) | 58.0 | 67.4 | 103.2 |
> > >
> > >
> > > Once again, we are deeply grateful for your detailed review and invaluable suggestions.

---

### Official Review · Reviewer_GCCs · 2023-07-26
**A good work on benchmarking the semiconductor materials. But the reviewer is not an expert in the field, hence, it is an outsider's perspective on MLFF benchmark.**

**Rating:** 8
**Confidence:** 2
**Clarity:** Most part is clear, but some part sti…

**Strengths:**

•	High-quality datasets for silicon nitride and hafnium oxide semiconductor materials generated through extensive, costly simulations. This provides realistic and complex data for benchmarking MLFF models.
•	Comprehensive benchmarking of 10 diverse existing machine learning approaches for modeling atomic-level material properties, spanning both traditional methods and graph neural networks. Testing a wide range of models gives greater insights.
•	Introduction of simulation-based performance metrics beyond standard force and energy prediction errors. Using indicators such as radial distribution functions and bulk modulus provides a more rigorous assessment of real-world physical accuracy and utility.
•	In-depth comparative analysis and discussion examining the tradeoffs between accuracy, inference speed, and physical consistency exhibited by different models. This provides valuable guidelines for selecting appropriate models for different use cases.


**Additional Feedback:**

No

**Correctness:**

The claims made in the paper appear to be correct. The datasets are generated using density functional theory (DFT) simulations under various conditions and scenarios. This is a standard approach for creating reference datasets in materials science.

**Documentation:**

The documentation appears to be complete with sufficient details on download and use the dataset, along with the licensing.

**Limitations:**

Yes, limitations and potential societal impact of the work have been adequately addressed in the Appendix.

**Opportunities For Improvement:**

•	The motivation and importance of some of the specific material properties and simulation indicators used for benchmarking could be explained more clearly for readers unfamiliar with semiconductor physics. For example, elaborating on why properties like bulk modulus are crucial for performance.
•	Inclusion of more visualizations and examples demonstrating model prediction behaviors could improve understanding for non-specialists. For instance, showing how radial distribution functions vary between different models.

**Relation To Prior Work:**

Prior MLFF benchmarks are appropriately cited such as ANI, rMD17, COLL. How this work differs from these existing benchmarks in the way of allowing evaluation of material properties beyond energy, force, and stress is also clearly explained.

**Summary And Contributions:**

This paper introduces new datasets, evaluation metrics and analysis focused on machine learning force fields for semiconductor materials. The goal of MLFF is to predict atomic-level properties of semiconductor materials.  The authors used expensive simulations to generate datasets for two important semiconductor materials - silicon nitride and hafnium oxide. A range of MLFF models are benchmarked, spanning traditional feature engineering methods and graph neural networks. The proposed simulation-based metrics provide insights beyond standard force and energy prediction errors. However, clearer justification for the specific indicators chosen would be helpful for newcomers. A key contribution seems to be proposing new testing methods beyond normal error metrics. The results show tradeoffs between accuracy, speed, and physical realism for different models.   The benchmarks and analysis appear valuable for researchers developing MLFF models. The new testing methods help identify models that could be useful in real applications. I appreciated the authors' efforts to explain concepts and motivation in a layman friendly manner. Overall, this appears to be a high-quality effort to generate robust benchmarks and critically analyze MLFF models for an important domain. As an outsider, I believe this work could significantly benefit semiconductor simulation and MLFF research.

---

> ### Author Response · Authors · 2023-08-18
> **Authors' Response to Reviewer GCCs**
>
> We are grateful for your valuable suggestions aimed at improving the quality of our paper. In response, we have integrated your recommendations as outlined below, with the intention of further enhancing the quality of our work.
>
> > The motivation and importance of some of the specific material properties and simulation indicators used for benchmarking could be explained more clearly for readers unfamiliar with semiconductor physics. For example, elaborating on why properties like bulk modulus are crucial for performance.
>
> * We have revised our paper to enhance clarity on the indicators, highlighting their significance and the underlying logic of material metrics. Acknowledging that some readers might not be well-versed in semiconductor physics and atomic simulations, we have incorporated dedicated sections in the manuscript (**Section D.1.1, D.2.1**, and **D.3.1**). These additional sections aim to simplify the topic by presenting clear examples, ensuring a broader understanding, especially within the ML community. We believe this will facilitate a better grasp of the discussed indicators.
>
> > Inclusion of more visualizations and examples demonstrating model prediction behaviors could improve understanding for non-specialists. For instance, showing how radial distribution functions vary between different models.
>
> * In response to the reviewer's suggestion, we have incorporated visual representations of RDFs (**Figures D.2** and **D.3**) alongside visualized equation of state (EoS) plots (**Figures D.4** and **D.5**) into the Appendix D. These additions serve dual purposes: they not only provide a detailed comparative overview of the models' prediction behaviors but also enhance understanding for readers who might not be deeply familiar with the domain. Notably, these plots allow for a clear visual distinction between the models' performance under in-distribution and out-of-distribution scenarios, offering further insight into their predictive capabilities. The RDF and EoS results for all the conditions and the structures presented in the paper are voluminous. Instead of including them all in the appendix, these results will be provided in a zip file as part of our supplementary material. We believe these modifications align with the intent behind your feedback and elevate the clarity of our paper.
>
> In conclusion, we would like to express sincere gratitude to the reviewers for their insightful suggestions and valuable comments. Your feedback has been instrumental in refining the quality and rigor of this paper. I believe that the revisions have addressed your concerns and have greatly enhanced the overall clarity and robustness of the work. Thank you for your time and effort in guiding this research to its best possible form.

---

### Author Response · Authors · 2023-07-12

We deeply apologize for a typo in a script file which we submitted.
As a solution, in scripts/preprocess_data/run_preprocess_data.sh, "maaxneigh" should be fixed as "maxneigh".

---

### Decision · Program_Chairs · 2023-09-22

**Decision:**

Accept (Poster)

**Comment:**

This paper introduces a new benchmark for evaluating machine learning force fields for semiconductor simulations, consisting of a dataset of two semiconductor materials (HfO, SiN) with DFT calculations performed with PBE, training and evaluation (ID and OOD) splits, several evaluation metrics as well as results for 10 baseline models. I appreciate the thoroughness in the empirical evaluations of baseline models. One concern is that in its current form, the dataset is limited in that it only considers two materials — the authors acknowledge that they plan to add Hf-Zr-O and Hf-Si-O in the near-term and more diverse compositions in future. Once that happens, this dataset has the potential to serve as a strong benchmark for AI researchers working on GNNs for atomistic modeling in particular and AI for science more broadly. Regardless, I agree with the reviewers that the paper in its current form (alongside boilerplate open-source code) is a good contribution to spur research in this important area. I encourage the authors to incorporate the reviewers' suggestions in the main paper. I recommend acceptance.